# SNAP: Self-Supervised Neural Maps for Visual Positioning and Semantic Understanding

**Paul-Edouard Sarlin**[1][*]
psarlin.com

**Eduard Trulls**[2]
trulls@google.com

**Marc Pollefeys**[1]
marc.pollefeys@ethz.ch

**Jan Hosang**[2]
hosang@google.com

**Simon Lynen**[2]
slynen@google.com

[1]ETH Zurich     [2]Google Research

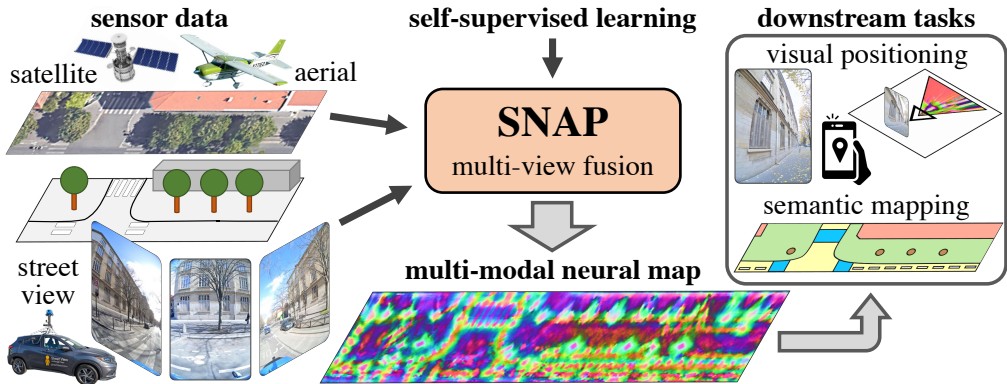

Figure 1: We learn *neural 2D maps* from multi-modal imagery using camera poses. SNAP outperforms the state of the art in visual positioning, and by solving localization as a proxy task learns easily interpretable, high-level semantics through self-supervision alone, without any semantic cues.

## Abstract

Semantic 2D maps are commonly used by humans and machines for navigation purposes, whether it's walking or driving. However, these maps have limitations: they lack detail, often contain inaccuracies, and are difficult to create and maintain, especially in an automated fashion. Can we use *raw imagery* to automatically create *better maps* that can be easily interpreted by both humans and machines? We introduce SNAP, a deep network that learns rich *neural* 2D maps from ground-level and overhead images. We train our model to align neural maps estimated from different inputs, supervised only with camera poses over tens of millions of StreetView images. SNAP can resolve the location of challenging image queries beyond the reach of traditional methods, outperforming the state of the art in localization by a large margin. Moreover, our neural maps encode not only geometry and appearance but also high-level semantics, discovered without explicit supervision. This enables effective pre-training for data-efficient semantic scene understanding, with the potential to unlock cost-efficient creation of more detailed maps.[†]

## 1 Introduction

Semantic 2D maps such as Google Maps are ubiquitous in our daily lives, used by billions of people. They offer compact, yet easily interpretable representations of the world from a bird's-eye view, allowing us to effectively navigate large outdoor environments by foot or vehicle. By contrast,

---

[*]Work done during an internship at Google.     [†]Code available at github.com/google-research/snap

37th Conference on Neural Information Processing Systems (NeurIPS 2023).

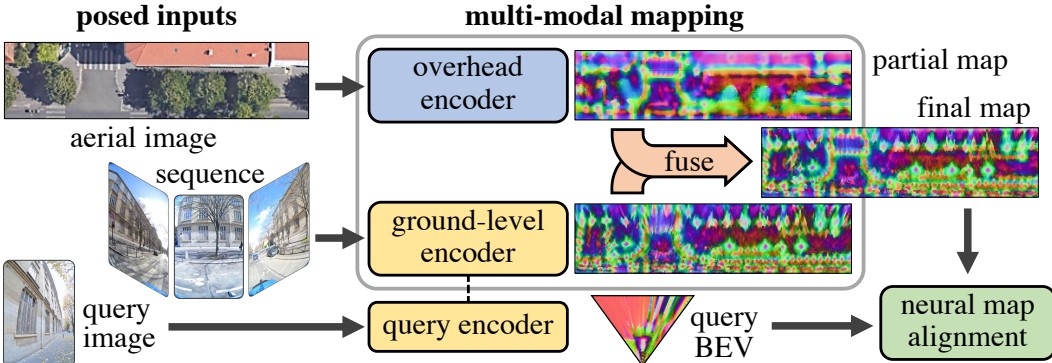

Figure 2: **Training architecture.** We feed overhead and ground-level imagery to per-stream encoders (Sec. 2.1) to produce 2D bird's-eye-view neural maps, fused via cell-wise max-pooling (Sec. 2.2). We also extract a 'query' neural map from a single ground-level image with the same ground-level encoder. Given known poses, we train SNAP by simply registering 'query' and 'scene' maps (Sec. 3).

machines position themselves in the real world through computer vision, which remains dominated by structure-based approaches [79, 83, 56, 74, 39, 65] relying on basic hand-crafted [54, 4] or learned [58, 111, 23, 69, 101, 75, 53] primitives, such as points or lines. These approaches build 3D maps with Structure-from-Motion (SfM) and then localize query images via 2D-3D registration. Their complexity and many components (feature extraction and matching, bundle adjustment, pose refinement, etc.) make it difficult to tune [42] or update [25] them, and to learn high-level priors end-to-end [10, 8, 76]. They are also costly to store and generally not reusable for other applications.

Recent works such as OrienterNet [78] instead learn planar, neural representations from the same 2D semantic maps that humans use. These maps encode scene geometry and semantics and can be used for visual positioning with sub-meter accuracy. This approach is however limited to a few semantic classes, and the maps it is based on can be inaccurate, costly to obtain, and difficult to maintain.

We argue that maps are most useful for figuring out where we are when they are *abstract* enough to be robust to temporal changes, yet preserve enough *geometric and semantic information* to yield high-quality correspondences with the physical world. Our work, SNAP, shows that, by learning 2D neural maps for localization, meaningful semantics emerge without explicitly supervising them. These semantics improve positioning accuracy and also make our maps usable for other tasks (Fig. 1).

SNAP leverages the complementary strengths of different input modalities, like ground-level and overhead imagery, by fusing them into a single 2D neural map (Fig. 2). It can flexibly and efficiently integrate arbitrary combinations of data captured at different points in time, which is key to continuously update maps in a changing world. We train it end-to-end to estimate the pose of a query image relative to the mapping images, by simply aligning their neural maps. This kind of contrastive learning requires only sensor poses, which can be easily obtained with photogrammetry [45, 37]. We train and evaluate SNAP on a dataset with 50M StreetView images[‡] from 5 continents, *orders-of-magnitude* larger and more diverse than comparable academic benchmarks.

Despite training only for a positioning objective, we observe that our neural maps learn easily-interpretable, high-level semantics without the need for explicit semantic cues (Fig. 3), and demonstrate that they provide an effective pretraining for semantic understanding tasks by fine-tuning them on little labeled data. This can potentially unlock cost-efficient creation of more detailed and richer maps, readable by humans and machines alike, while providing state-of-the-art visual positioning.

Our main contributions are as follows. (i) We introduce a simple and lightweight encoder to estimate bird's-eye view maps from ground-level imagery, combining principles from multi-view geometry with strong monocular cues. (ii) We fuse different imaging modalities to integrate and benefit from complementary cues. (iii) We show how to train our model by aligning neural maps in a contrastive learning framework, using RANSAC to mine hard negatives. (iv) We outperform the state of the art on visual positioning and register image queries beyond the reach of traditional methods (Fig. 3). (v) We demonstrate that high-level semantics emerge by learning to align neural maps, without any explicit supervision, and fine-tune them on semantic understanding with few labels (Fig. 6).

---

[‡]Analytical use of StreetView imagery was done with special permission from Google.

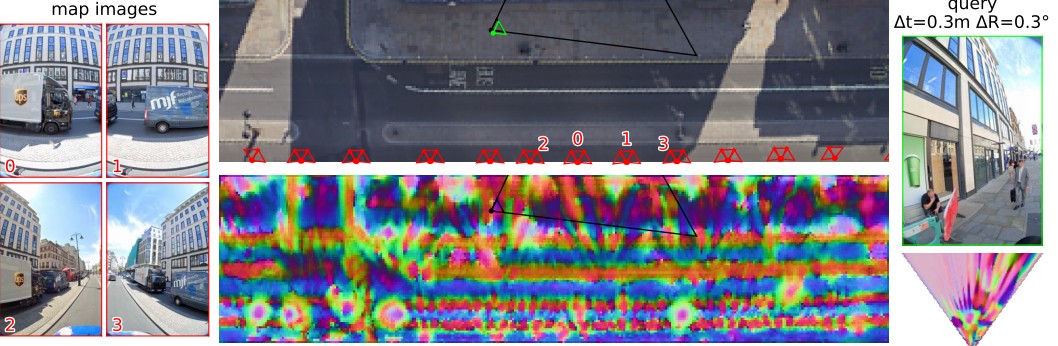

a) A facade with repeated structure and large viewpoint differences between query and map images (medium).

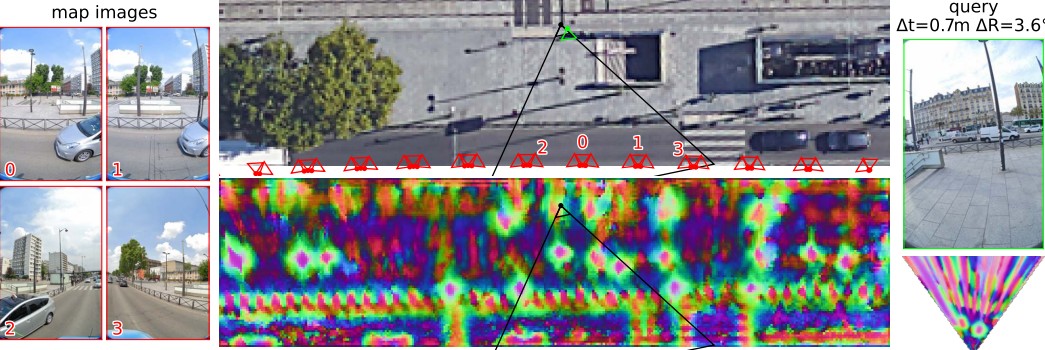

b) Opposite views, localized using the ground and poles, observable in the neural map (hard).

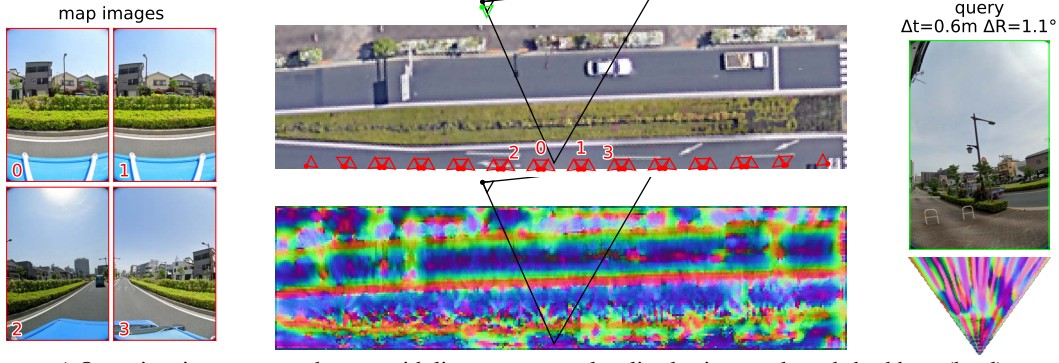

c) Opposite views on a road scene with linear structure, localized using a pole and shrubbery (hard).

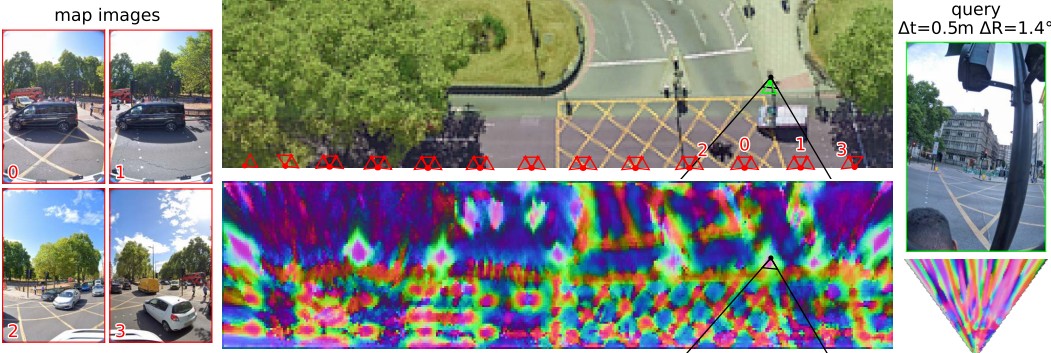

d) Opposite views, localized using a close-up of a pole and the markings on the ground (hard).

Figure 3: **Single-image localization.** We show the 3-DoF poses of map images **(red)**, the GT query pose **(black)**, and the pose predicted by SNAP **(green)** with its error $(\Delta t, \Delta R)$. SNAP can estimate accurate poses even for extreme opposite-views. We visualize neural maps by projecting them to RGB using PCA. Notice how objects like trees, poles, curbs or road markings are clearly recognizable.

## 2 Mapping the world with neural maps

We now formalize neural maps, and describe a neural network architecture to infer them from raw sensor data. Our goal is to infer a more generic neural representation that can encode both the geometry, semantics, and appearance of a given point in the 2D world.

**Problem formulation:** For a 3D scene, such as a large outdoor environment, we consider a local, 3D Cartesian coordinate system such that the $z$ axis points upwards along the gravity direction. A neural map $\mathbf{M}$ is defined over a regular grid that partitions the $xy$ plane into $I{\times}J$ square cells of size $\Delta$. Each cell $(i, j)$ is associated with a $D$-dimensional feature $\mathbf{M}_{ij} \in \mathbb{R}^D$. To infer such neural map, we leverage large quantities of raw imagery captured by diverse cameras.

**Input modalities:** Ground-level images are captured by cameras mounted on StreetView cars or backpacks [18]. They are often part of a sequence of multi-camera frames. As such, they are very unevenly distributed throughout space. Each image offers a high resolution view of a small area, mainly limited by the occlusion of static or dynamic objects like buildings or vehicles. On the other hand, overhead images are captured by cameras mounted on planes or satellites. These images benefit from high spatial coverage at a uniform but low resolution. Their visibility is mostly affected by vertical occluders like trees. Ground-level and overhead images capture different aspects of the environment and are thus complementary.

**Assumptions:** All images of either modality are calibrated and registered with respect to the map coordinate system. Each image $n$ follows a projection function $\Pi_n : \mathbb{R}^3 \to \mathbb{R}^2$ that maps a 3D point in the world to a 2D point on the image plane. $\Pi_n$ combines the camera pose ${}_w\mathbf{T}_n \in \mathrm{SE}(3)$ and the camera calibration, including lens distortions. Overhead images are ortho-rectified, such that world points along the $z$ axis project onto the same pixel coordinate. As this process relies on a coarse digital surface model [27], fine details like poles are not rectified and may result in artifacts, which SNAP can however learn to account for.

### 2.1 Fusing multi-modal representations

Each location in the world is observed by an arbitrary number of images for each modality, captured at arbitrary points in time. We thus follow a late-fusion strategy that first encodes each modality separately and only finally fuses them (Fig. 2). This can flexibly adapt to the available inputs and efficiently handle arbitrary spatial distributions of data.

**Encoding:** We design two encoders that each combine a subset of observations $n$ into a single-modality neural map $\mathbf{M}^n$ defined over the same grid as $\mathbf{M}$. $\Phi_{\mathrm{OV}}$ encodes a single tile of overhead orthoimagery, while $\Phi_{\mathrm{SV}}$ encodes a single image or multiple covisible ground-level StreetView images, $e.g.$, a multi-view sequence. To best resolve the 3D information from perspective shots at arbitrary viewpoints, $\Phi_{\mathrm{SV}}$ leverages both multi-view observations and monocular cues. We describe its architecture in detail in Sec. 2.2. $\Phi_{\mathrm{OV}}$, on the other hand, is a simple U-Net-style CNN [71] that computes a feature for each pixel of the overhead orthoimage, which is then resampled into the grid.

**Fusion:** We obtain the final neural map by fusing the set of encoded maps $\{\mathbf{M}^n\}$ using a cell-wise max-pooling operation, $i.e.$, $\mathbf{M}_{ij} = \max_n \mathbf{M}_{ij}^n \; \forall \; (i, j) \in I{\times}J$. This can combine maps with different spatial extents, which is essential to scale to large areas. The *max* aggregation picks the best estimate among all inputs for each feature channel and thus handles partial observations, such as when the road surface cannot be resolved in overhead images because it is occluded by trees.

### 2.2 Ground-level image encoder

We design a single module, $\Phi_{\mathrm{SV}}$, that can arbitrarily encode one or multiple images, ordered or not. $\Phi_{\mathrm{SV}}$ first fuses the image data into 3D space and later projects it vertically into the map plane (Fig. 4). This design can handle arbitrary ground geometries and accurately resolve the 2D location of overhanging 3D structures, like street lights. The 3D fusion leverages both multi-view geometry and strong monocular cues learned end-to-end. $\Phi_{\mathrm{SV}}$ can thus resolve objects that are observed by a single image, while maximizing accuracy when multiple observations are available.

**Monocular inference:** We consider an unordered set of $N$ images $\{\mathbf{I}^n\}$, $N{\geq}1$. Each image $n$ is encoded independently by a CNN $\Phi_{\mathbf{I}}$ into a $C$-dimensional feature image $\mathbf{F}^n \in \mathbb{R}^{H \times W \times C}$. $\Phi_{\mathbf{I}}$ also estimates a pixel-wise depth $\mathbf{S}^n \in \mathbb{R}^{H \times W \times D}$ as a score over $D$ depth planes along the ray of each

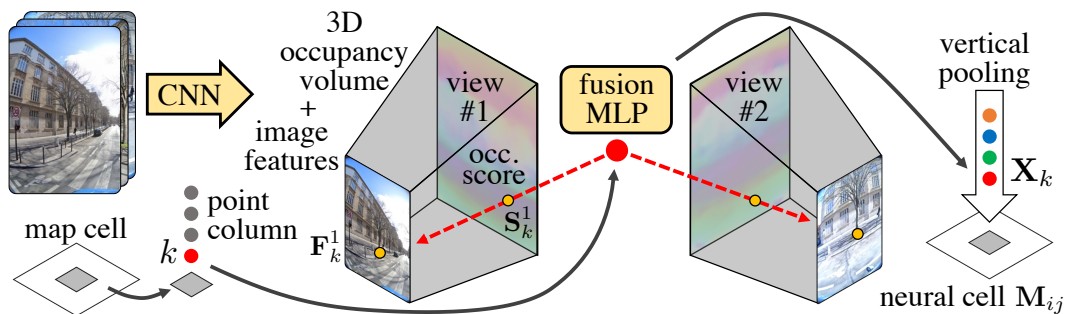

Figure 4: **Ground-level encoder: combining multi-view geometry and monocular priors.** We use a CNN to predict pixel-wise features and a monocular occupancy volume, separately for each view. We then interpolate them over a column of 3D points (at predefined heights), for each 2D cell. Finally, a simple MLP combines them into features $\mathbf{X}_k$ that are pooled along the column, into a neural cell.

pixel. $\mathbf{S}^n$ is similar to a frustum-aligned occupancy volume [78, 67] but contains unnormalized logits of a depth distribution. Instead of regressing a single value, this encodes the full depth uncertainty along the ray and thus allows $\Phi_{\mathbf{I}}$ to provide meaningful multi-modal estimates. We distribute the depth planes uniformly in log space to correlate with the uncertainty of monocular depth estimation [3, 78].

**Multi-view fusion:** To fuse information in 3D, we define $K$ horizontal planes at heights $\{z_k\}$, which are uniformly distributed within a range of interest defined with respect to the height of the camera [85, 50], *e.g.*, from 4 m below to 8 m above. For a 2D map cell $(i, j) \in I \times J$, we consider its center point $(x, y)$ and a column of 3D points $\{\mathbf{P}_k = (x, y, z_k)\}$. For each 3D point $k$, we define the subset of views that best observe it as $\mathcal{N}_k \subseteq \{1 \ldots N\}$, *e.g.*, those that are closest spatially. We project the point to each of these views, obtain a 2D observation $\mathbf{p}_k^n = \Pi_n(\mathbf{P}_k)$, and sample the corresponding feature image with bi-linear interpolation: $\mathbf{F}_k^n = \mathbf{F}^n[\mathbf{p}_k^n]$. Given the depth $d_k^n$ of $\mathbf{P}_k$ in the corresponding view, we also tri-linearly interpolate a score from the depth prior: $\mathbf{S}_k^n = \mathbf{S}^n[\mathbf{p}_k^n, d_k^n]$. Intuitively, $\mathbf{S}_k^n$ is low if the 3D point is in free space or is occluded in view $n$. Following common practice in learned multi-view stereo [110, 106], we then compute feature consistency statistics, as mean and variance $(\boldsymbol{\mu}_k, \boldsymbol{\sigma}_k) \in \mathbb{R}^C$, weighted by the depth priors:

$$\boldsymbol{\mu}_k = \sum_{n \in \mathcal{N}_k} w_k^n \, \mathbf{F}_k^n \quad \text{and} \quad \boldsymbol{\sigma}_k = \sum_{n \in \mathcal{N}_k} w_k^n \, (\mathbf{F}_k^n - \boldsymbol{\mu}_k)^2 \quad \text{with} \quad w_k^n = \operatorname*{softmax}_{n \in \mathcal{N}_k} \mathbf{S}_k^n \ . \quad (1)$$

A Multi-Layer Perceptron (MLP) fuses this information into a feature $\mathbf{X}_k$, which is finally pooled across all points in the column, resulting in a neural map cell $\mathbf{M}_{ij}$:

$$\mathbf{M}_{ij} = \max_k \mathbf{X}_k \quad \text{with} \quad \mathbf{X}_k = \text{MLP}\left(\left[\boldsymbol{\mu}_k, \, \boldsymbol{\sigma}_k, \, \max_{n \in \mathcal{N}_k} \mathbf{S}_k^n\right]\right) \ . \quad (2)$$

Adding the maximum depth score differentiates free and occupied space when the point is observed by a single image. This makes it possible to use the same model for single images and sequences.

By tightly combining 3D geometry and representation learning, our approach leverages both monocular priors and multi-view information, while past research on 2D mapping or 3D reconstruction typically relies on only one of the two. Compared to expensive Transformers [102] or 3D CNNs [33], we show that a simpler, lightweight MLP is effective at fusing multi-view information, inspired by [82]. Compared to top-down 2D CNNs that squash the vertical dimension [34, 70], this MLP is more expressive and makes our neural maps equivariant to 2D translations and rotations and invariant to translations along the vertical axis. Overall, this simple design enables scaling to very large scenes, which is critical to provide hard negatives for contrastive learning and ultimately learn rich semantics.

## 3 Learning from pose supervision

**Alignment as contrastive learning:** We want neural maps to encode high-level semantic information about the environment. Given recent advances in self-supervised learning [14, 64], we hypothesize that this can emerge from learning distinctive features that distinguish one location from another and that are invariant to viewpoint and temporal appearance changes. Intuitively, *good maps help us identify where we are*. More generally, good maps are such that we can unambiguously align them

when inferred from partial inputs. Consider neural maps $\mathbf{M}^Q$ and $\mathbf{M}^R$ obtained from two disjoint subsets of inputs, the query $Q$ and the reference $R$. In camera pose estimation, $Q$ corresponds to a single ground-level image and $R$ to a sequence of images with an aerial tile. Because our encoder is flexible, we can use the same shared model to encode $Q$ and $R$ (Fig. 2). $\mathbf{M}^Q$ is defined over a grid $\mathbf{G}^Q \in \mathbb{R}^{I \times J \times 2}$ in a local coordinate frame, *e.g.*, aligned with the query camera, where $\mathbf{G}^Q_{ij}$ is the center point of cell $(i, j)$, while $\mathbf{M}^R$ is defined in the world frame.

We define a score function $E(\mathbf{T}; \mathbf{M}^Q, \mathbf{M}^R) : \mathrm{SE}(2) \to \mathbb{R}$ that evaluates the consistency between $\mathbf{M}^Q$ and $\mathbf{M}^R$ given an estimate of their 3-DoF relative pose $_R\mathbf{T}_Q \in \mathrm{SE}(2)$. To distinguish the ground-truth pose $_R\mathbf{T}^*_Q$ from $K$ other, incorrect poses $\{_R\mathbf{T}^k_Q\}$, we want to increase $E(_R\mathbf{T}^*_Q)$ and decrease $E(_R\mathbf{T}^k_Q)$ (omitting $\mathbf{M}^Q$ and $\mathbf{M}^R$ for brevity). This corresponds to a contrastive learning problem, for which we minimize the InfoNCE loss [63]

$$\mathrm{Loss}\left(\mathbf{M}^Q, \mathbf{M}^R\right) = -\log \frac{\exp\left(E\left(_R\mathbf{T}^*_Q\right)/\tau\right)}{\sum_{k \in \{*, 1...K\}} \exp\left(E\left(_R\mathbf{T}^k_Q\right)/\tau\right)} \quad, \tag{3}$$

where $\tau$ is a learnable temperature parameter. Neural maps are trained end-to-end and require only relative poses $_R\mathbf{T}^*_Q$, which can be easily obtained at a large scale using photogrammetry [45, 37].

**Featuremetric pose scoring:** A linear layer projects each neural map $\mathbf{M}$ to a lower-dimensional, L2-normalized map $\bar{\mathbf{M}}$. This creates an information bottleneck that encourages compact features. The score $E$ evaluates the consistency of two neural maps as the similarity of each cell after warping:

$$E(_R\mathbf{T}_Q) = \frac{1}{IJ} \sum_{(i,j) \in I \times J} \max\left(\bar{\mathbf{M}}^{Q\top}_{ij} \bar{\mathbf{M}}^R \left[_R\mathbf{T}_Q \cdot \mathbf{G}^Q_{ij}\right], 0\right) \quad, \tag{4}$$

where $_R\mathbf{T}_Q$ transforms a grid point from coordinate frames $Q$ to $R$ and $[\cdot]$ interpolates the map at this location. $\max$ clips negative scores to zero to reduce the impact of outliers, as in robust optimization.

**Negative sampling:** A critical and well-studied aspect of contrastive learning is the selection of negative samples [35, 98, 108]. Hard negatives should be high-likelihood but incorrect predictions, so as to push the probability mass to the ground truth. Random poses can be easily distinguished and exhaustive voting in the 3-DoF pose space is computationally infeasible at high resolution [78, 6, 28]. Instead, we use RANSAC [29] to sample poses that are consistent with the predicted features. We sample pairs of 2D-2D correspondences between all cells of both neural maps and solve for the relative pose using the Kabsch algorithm [43]. Inspired by PROSAC [19], we sample a correspondence between cells $(i, j)$ and $(k, l)$ based on its feature similarity with probability $P_{ijkl} = \underset{ijkl}{\mathrm{softmax}}\left(\bar{\mathbf{M}}^{Q\top}_{ij} \bar{\mathbf{M}}^R_{kl}/\tau\right)$. Unlike NG-RANSAC [10], gradients are propagated through the scoring rather than the sampling and are thus much smoother. Because the sampling and scoring mirror similar featuremetric errors, negative samples become harder as the learning proceeds.

**Inference-time alignment:** SNAP can estimate the unknown 3-DoF relative pose between any two neural maps. We estimate each map in the sensor coordinate frame, establish tentative correspondences by matching their cells, sample pose hypotheses, and select the pose with the highest score. This includes single-image positioning, where the query map $\mathbf{M}^Q$ covers the camera frustum. The vertical pooling requires that the gravity direction is known, which is a reasonable assumption for applications like Augmented Reality (AR) and robotics [56, 113, 78]. Our framework also applies more generally to aligning any pair of inputs, including sequence-to-sequence and aerial-to-ground registration, which is required in the first place to pose mapping data in a common reference frame.

## 4 Related work

**Visual positioning** is most commonly tackled with geometric approaches [80, 40, 74] that rely on point correspondences across images and sparse 3D point clouds built with SfM [83, 2]. They then estimate the 6-DoF query pose with a robust solver [29, 20, 21, 19, 15, 5] from correspondences with the reference model or images. Such correspondences are most often estimated by sparse local features [54, 4]. This process is complex and end-to-end back-propagation is impractical [8]. Past works have thus focused on learning specific components like feature extraction [111, 58, 23, 24,

99, 69, 26, 101, 105, 55], matching [112, 115, 75, 91, 41, 117, 107, 53], and pose [104, 76] or point cloud refinement [52]. Coarse GPS location and gravity direction are commonly assumed to be known [113, 56, 93]. In AR and robotics, the height of the camera can be estimated as the distance to the ground in a local SLAM reconstruction [78]. These assumptions reduce the problem to 3-DoF estimation and make it more amenable to end-to-end learning. MapNet [38] also learns end-to-end 3-DoF visual mapping and localization but requires sequences of depth inputs. Recent works leverage overhead instead of ground-level images [87, 86, 109, 28]. They easily scale to large scenes but only in open-sky areas. Their accuracy is also limited by the low resolution of aerial imagery. Our work combines the strengths of both ground-level and overhead imagery by learning end-to-end how to best fuse them for 3-DoF positioning. Our differentiable pose estimation, based on RANSAC, is more efficient [38, 28, 78], robust [87], and stable [10, 8] than previous approaches.

**Semantic representations** can largely benefit loop closure [84] and pose estimation [100]. OrienterNet [78] learns 3-DoF positioning end-to-end from public 2D semantic maps that are more compact yet detailed enough for localization. Its accuracy is however limited because these maps have low spatial accuracy and are infrequently updated. It is also also restricted to few, explicit semantic classes that are often not discriminative. Differently, [49] learns finer-grained semantic classes for temporal and viewpoint consistency. Our work instead learns *implicit* semantics from posed imagery by combining end-to-end self-supervised learning with large amounts of data. This boosts the positioning accuracy and is an effective pre-training for semantic tasks.

**Neural scene representation** is an active topic of research. MLPs [57, 96] and tokens [73] are compact but lack geometric inductive bias. 3D voxel grids are more expressive and thus popular for reconstruction [66, 60, 92, 9, 119], rendering [59, 95], and semantic perception [17, 102, 13, 7] but are expensive to store and thus often restricted to small scenes. 2D grids, or Bird's-Eye Views (BEV), are more compact and thus scale to larger outdoor scenes by compressing the information along the vertical axis. Neural BEVs can be learned from images for supervised semantic tasks [72, 50, 34, 67, 70], 3D reconstruction [66], self-supervised view synthesis [85], and 3-DoF positioning [78, 28, 38]. These approaches assume planar scenes or rely on monocular priors only, even if multiple views are available. Instead, we combine these priors with multi-view fusion [110, 82, 106] to leverage information from image sequences and better resolve objects in large scenes.

**Self-supervised learning** leverages unlabeled datasets to learn representations useful for down-stream tasks. Many works focus on image- or pixel-level contrastive learning for semantic tasks [63, 35, 14, 62, 36]. View synthesis from few images typically learns lower-level representations [57, 85]. Some works [49, 89] learn features for image matching across appearance changes. CoCoNets [48] learns representations for 3D scenes but requires perfect, synthetic depth maps. We learn high-level contrastive scene representations from posed images and show that it translates to semantic mapping.

# 5 Experiments

**Data:** StreetView images are captured by rigs of 6 rolling-shutter cameras mounted on cars or on backpacks worn by pedestrians [18], which results in a wide diversity of viewpoints in street-level scenes. Multi-view 'frames' are captured synchronously every ∼5m. Sequences are captured between 2017 and 2022. We build mapping segments *only from car sequences* by partitioning each sequence into groups of 36 images that face either the left or right side of the road. We define each map grid as a $64 \times 16$ m tile aligned with the segment mid-frame, in which we render an aerial orthophoto with 20 cm ground sample distance. Query images are sampled from different sequences, captured from cars or backpacks, based on their frustum overlap, and are often taken years apart. We train with 2.5M segments and ∼50M queries from 11 cities across the world: Barcelona, London, Paris (Europe), New York, San Francisco (North America), Rio de Janeiro (South America), Manila, Singapore, Taipei, Tokyo (Asia), and Sydney (Oceania), reserving some areas in each city for validation. We test on 6 different cities (Amsterdam, Melbourne, Mexico City, Osaka, São Paulo, and Seattle), with 4k queries per city. This covers 5 continents, while academic localization benchmarks focus on tourism landmarks [42] or single cities in Europe or the US [81, 116, 77] – see details in Appendix D.

**Training and implementation:** In the ground-level encoder, $\Phi_\mathbf{I}$ is a U-Net [71] with a BiT ResNet backbone [46], pre-trained as in [114], and an FPN decoder [51], initialized randomly. We consider two models with different backbones: a 'large' R152x2 (353M parameters) and a 'small' R50x1 (84M parameters). $\Phi_{OV}$ is a similarly-defined R50x1+FPN. In multi-view fusion (Sec. 2.2) we use

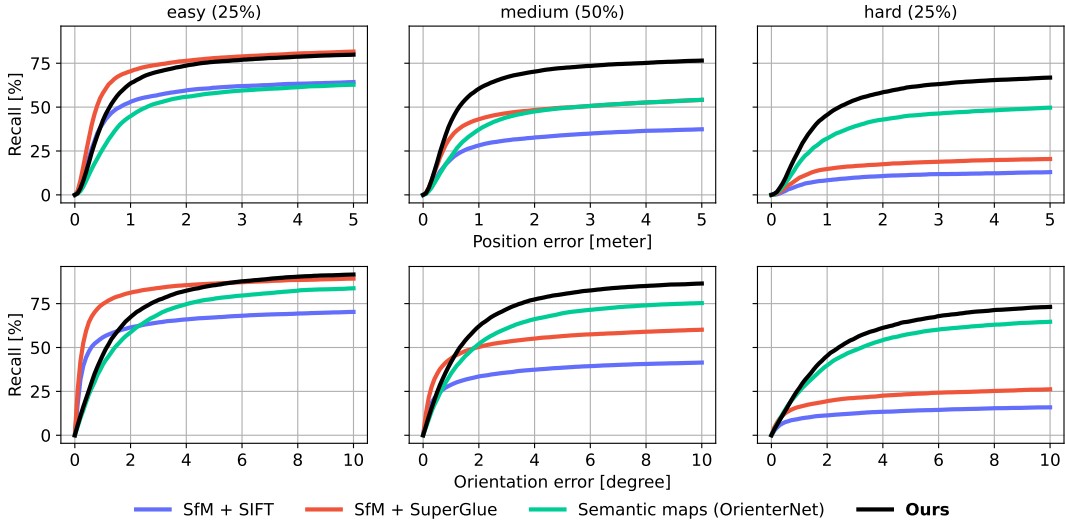

Figure 5: **Single-image positioning with different maps.** Localizing with our neural maps yields a higher recall than established approaches based on feature matching (*SfM + X*), especially for hard queries with low visual overlap. Neural maps are also more suitable for positioning than semantic maps because they encode richer and thus more discriminative information.

| | Algorithm | Inputs | Easy (25%) | Med. (50%) | Hard (25%) | All (100%) |
|---|---|---|---|---|---|---|
| SfM | + SIFT [54] | StreetView | 47.0 / 54.4 | 24.9 / 29.9 | 7.6 / 9.7 | 27.1 / 32.1 |
| | + SuperGlue [75] | StreetView | **63.0 / 71.1** | 38.0 / 44.4 | 13.1 / 16.1 | 39.2 / 45.2 |
| OrienterNet [78] | | semantic | 35.6 / 47.2 | 29.3 / 39.5 | 24.8 / 34.8 | 30.0 / 40.6 |
| **SNAP-large** | | multi-modal | 48.9 / 62.3 | **46.9 / 59.5** | **34.5 / 47.6** | **44.4 / 57.4** |
| **ResNet-152x2** | | StreetView | 45.8 / 58.4 | 43.9 / 56.0 | 29.5 / 41.7 | 41.0 / 53.2 |
| | | aerial | 27.4 / 40.6 | 25.3 / 37.5 | 20.8 / 32.3 | 24.8 / 37.1 |
| **SNAP-small** | | multi-modal | 45.2 / 59.0 | 41.9 / 54.8 | 29.6 / 42.0 | 39.9 / 52.9 |
| **ResNet-50** | | StreetView | 42.2 / 54.9 | 38.1 / 50.1 | 24.5 / 36.4 | 36.0 / 48.2 |
| | | aerial | 23.9 / 35.6 | 21.9 / 32.8 | 17.9 / 27.5 | 21.5 / 32.3 |

Table 1: **Single-image positioning.** We report the area under the recall curve (AUC) up to thresholds (2.5 m/5°) and (5 m/10°). Our large and small multi-modal models are more accurate than classical *SfM + X* approaches for medium and hard queries, which matter most in practical applications. Fusing both StreetView and aerial imagery is more accurate than using only one of them.

$D{=}32$ depth planes and $K{=}60$ height planes $\{z_k\}$ uniformly distributed within 12 m. Neural maps $\mathbf{M}$ and matching maps $\bar{\mathbf{M}}$ have dimensions 128 and 32, respectively, and are defined over $64{\times}16$ m grids with 20 cm ground sample distance. Query BEVs have a maximum depth of 16 m. At training time, neural maps are built from one aerial tile and one SV segment, with each of the two randomly dropped, similarly to dropout [90]. We use a subset of $N{=}20$ views, some of them at a $\pm60°$ angle, which we empirically found provides a good coverage/memory trade-off. See details in Appendix E.

**Visual positioning:** We build a map for each segment using all 36 views and evaluate the 3-DoF query pose in terms of position and orientation errors. While many academic benchmarks use much larger mapping areas, we argue that GPS and motion priors often make this unnecessary for practical applications [56]. We slice the results by difficulty in terms of query-scene overlap based on the distance between the query and its closest map view, in position $\Delta t$, and orientation $\Delta\theta$. We split in the data into 3 groups: 'easy' ($\Delta t{<}10$ m and $\Delta\theta{<}45°$, ~25% of the data), 'hard' ($\Delta t{>}10$ m and $\Delta\theta{>}60°$, ~25%), and 'medium' (the remaining ~50%). We compare our approach to hloc [74], a state-of-the-art [81, 77] structure-based 6-DoF localization system based on COLMAP [83], a popular Structure-from-Motion framework, with correspondences estimated by either RootSIFT [54, 4] or SuperPoint+SuperGlue [23, 75], a learned feature and matcher. Note that these approaches can only leverage ground-level imagery. We match the query to all map images, without using hloc's

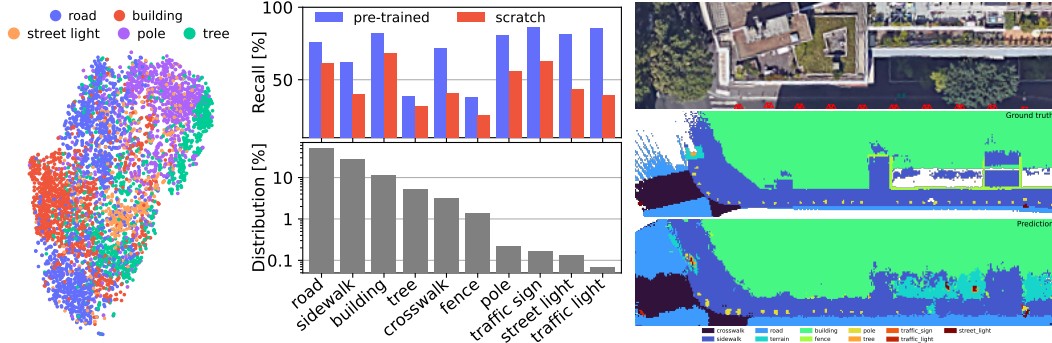

Figure 6: **2D Semantic mapping.** Left: t-SNE visualization of the neural map features learned by SNAP, colored by their ground-truth semantic class. SNAP discovers different categories of objects common in outdoor urban scenes, which yields clearly distinguishable clusters. Middle: Given a small labeled dataset, training a tiny CNN classifier to predict such classes from pre-trained features is more effective than training the entire SNAP model from scratch, especially for small and infrequent objects. Right: Test example with ground truth raster (middle) and prediction of the CNN (bottom).

retrieval component, and estimate the query's pose using RANSAC and a P2P solver with gravity constraint [94]. We evaluate the 6-DoF pose projected to 3-DoF. We also evaluate OrienterNet [78], which matches a query BEV with a semantic map. We re-implement and train it on overhead semantic rasters derived from SfM points with semantic labels obtained by fusing 2D image segmentations. Note that OrienterNet was originally trained on OpenStreetMap, which has limited coverage of small objects. While our rasters are noisy, they provide a consistent, global coverage of fine-grained classes like tree, streetlight, poles, etc. We checked our implementation with the authors. We also evaluate versions of SNAP trained with only either ground-level or aerial imagery – the latter is an extreme case of cross-view localization, similar to [28].

Fig. 5 and Tab. 1 show that SNAP outperforms the state of the art, COLMAP with Super-Point+SuperGlue, by a large margin: 25% relative. Structure-based approaches are more accurate for easy queries but significantly worse for hard ones (Fig. 3). Using ground-level imagery is crucial in most localization scenarios and performs ~46% relative better than using aerial imagery, whereas our multi-modal model performs ~8% relative better than the StreetView-only variant. We justify our design decisions using an ablation study in Appendix A. We report detailed results per city and per sequence type in Appendix C. Our framework is also efficient, as mapping takes 223 ms per segment and 6 ms per aerial tile, estimating a query BEV takes 14 ms and localizing it takes 86 ms, on an A100 GPU. In comparison, matching with SuperGlue takes 100ms per pair for 36 pairs per query, and is thus 36 times slower. Each tile of our matching maps has size 1.6 MB in fp16, while storing SuperPoint descriptors requires 5.3 MB on average.

**Semantic mapping:** We show that SNAP's neural maps are an effective pre-training for 2D semantic mapping. Existing approaches rely on ground truth 2D semantic rasters derived from the segmentation of LiDAR 3D point clouds. These are manually labeled, which is too expensive to generate enough data to train from scratch models that generalize across countries, sensors, seasons, and times of the day. Existing datasets [12, 7] thus rarely span more than a few cities and overfit supervised models to the local appearance. Instead, our self-supervised pre-training learns better features from a much larger dataset of posed imagery, which is much cheaper to acquire at scale. The information bottleneck forces SNAP to learn unified representations for objects, like street crossings or lights, that look very different across countries, and would require larger amounts of labeled data. Fig. 6-left shows a 2D t-SNE [103] visualization of SNAP's neural maps at points sampled on a few types of objects common in street scenes, according to their ground-truth semantic label. Points of the same class are clustered together. This clearly shows that neural maps learn to distinguish these objects without any semantic supervision, even if they are geometrically similar, *e.g.*, tree *vs* pole.

To evaluate the pre-training, we train a tiny CNN to predict semantic rasters from pre-trained neural maps, keeping SNAP frozen. We compare this to training the entire model from scratch (with the same backbones initialization [46, 114]). We derive 3k 64×16 m ground truth rasters from LiDAR point clouds captured by StreetView cars in 84 cities across the world. We train with 2k examples and report the recall of both approaches on 1k test examples in Fig. 6-center. Pre-training consistently

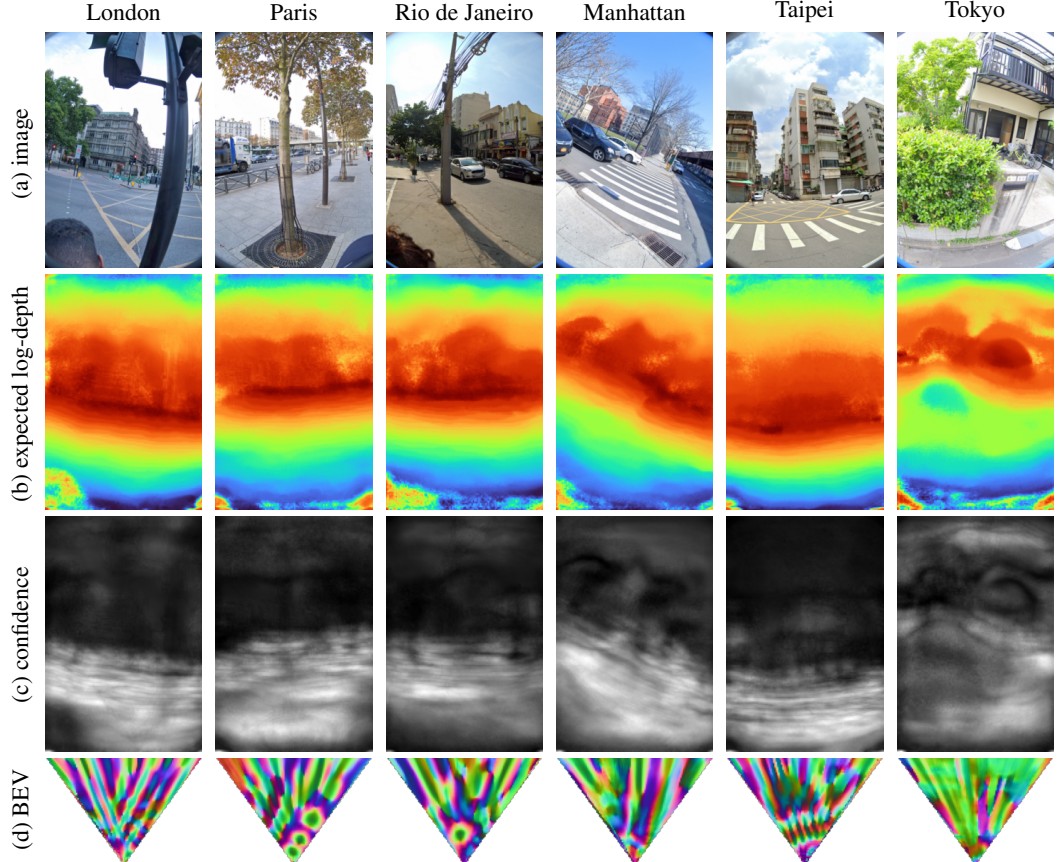

Figure 7: **Monocular depth priors learned by the ground-level encoder.** For each query image (a), we show: (b) the expected log-depth across all depth planes, from blue (close) to red (distant); (c) the total score along each ray $\log \sum_{i \in \{1...D\}} \exp \mathbf{S}[:,:,d_i]$, which reflects how useful or confident the prediction is; (d) the resulting bird's-eye view. The predictions are sensible for areas close to the ground and for lower parts of objects and buildings. Predictions in the sky and upper facades are not reliable because these areas are never covered by the height planes $\{z_k\}$ of the point columns.

yields better results for every class, with larger gains on more difficult/infrequent classes. While training from scratch massively overfits to such small dataset, our neural maps encode enough information to reach recalls over 70%. We show qualitative examples in Fig. 6-right and Appendix B.

**Monocular priors:** We visualize in Fig. 7 the occupancy predicted by SNAP as depth and confidence maps. SNAP learns sensible priors over the geometry of street scenes from only pose supervision.

**Limitations:** Our approach is not as accurate as structure-based methods given easy queries closer to map images (Fig. 5). We hypothesize this is partly due to operating at lower image resolutions. It also assumes gravity direction and a location prior, which are reasonable assumptions but restrict its use.

## 6  Conclusion

We present SNAP, a novel approach to build semantic, 2D neural maps from multi-modal inputs and train it by simply learning to align two neural maps in a contrastive framework. This simple objective yields a model that can localize queries beyond the reach of the state of the art in structure-based matching by discovering high-level semantics from self-supervision. Our neural maps are easily interpretable and provide an effective pre-training towards unlocking semantic understanding at scale.

**Broader impact:** This work has implications to privacy and surveillance. However, our 2D maps are too compact to preserve personal identifiable information, and likely more difficult to invert than point clouds [68], which despite ongoing efforts [88, 31, 32, 61, 118] remain susceptible to attacks [16].

**Acknowledgements:** We thank Bernhard Zeisl, Songyou Peng, Rémi Pautrat, and Michał Tyszkiewicz for their valuable feedback, Manuel Cabral and Johann Volz for providing useful code reviews, Tianqi Fan for helping processing the data, Lucas Beyer for providing the pre-trained ResNet backbones, Thomas Funkhouser and Kyle Genova for providing the ground truth semantic labels, and Arjun Karpur for helping run SuperGlue.

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

# Appendix

## A    Design decisions

In this section, we explain our design decisions and support them with an ablation study.

**Constraints:** Learning features that are discriminative requires sufficiently challenging negative pose samples. These arise from viewpoints that are visually similar to the ground truth viewpoint, for example due to repeated patterns, lack of distinctive features, or occlusions. In order to find sufficiently difficult negative samples in a training example, the map should be as large as possible. The size of the map and the number of mapping images are however limited by the amount of GPU memory available. We therefore found it critical to find the right balance between the size of the map, its spatial resolution, the number of views, and the batch size, as also reported in [34].

We found that a simple and lightweight model that saves memory is beneficial over a complex model that requires reducing the size of the map, the number of views, or the batch size. We thus favor explicit constraints by geometry (camera projection, 3D occupancy) rather than flexible but heavy mechanisms like attention or 3D convolutions. The effectiveness of our design shows that such complexity is not required. This enables the training of high-capacity models with maps of $64 \times 16$ m, at a resolution of 20 cm (~25k cells per map), from 20 images, and within 20 GB of GPU memory per example. Despite extensive memory optimizations, including the use of gradient checkpointing and mixed-precision training, using larger maps for training remains challenging.

**Ground-level encoder:** We consider four alternative designs for the ground-level encoder $\Phi_{\mathrm{SV}}$.

(a) **Fixed-height plane.** Instead of lifting the image information to 3D and pooling it vertically, the geometry of the scene can be approximated from a ground plane [1]. Driving applications commonly assume that this plane is horizontal and at a fixed height below the camera [86]. We train a variant that aggregates the multi-view information on a plane 2.5 m below the camera. This approach can hardly resolve the spatial location of overhanging structures, like street lights. It furthermore introduces distortions in the BEV if the image is not gravity aligned, which is the case for many capture platforms (such as StreetView backpacks or consumer phones), or the ground is not planar, as in many real environments.

(b) **No monocular occupancy.** Multi-view stereo [106, 110] typically does not explicitly leverage monocular geometry priors. We thus train a variant that omits the weighting by occupancy scores. Image features are thus painted identically along all depth planes for each ray and their mean and variance across all views is fed to the fusion MLP. This model can only leverage the bearing angles of point features (like poles) to disambiguate a pose but cannot resolve the location of line features.

(c) **No multi-view variance.** SimpleBEV [34] simply averages feature volumes obtained by painting image features along each ray. This corresponds to removing both variance and monocular priors from SNAP, making it even harder to resolve surfaces.

(d) **Ray conditioning.** Close to our approach, Sharma *et al.* [85] augment multi-view fusion with monocular cues, but do so by conditioning each ray feature by its 3D location, using an MLP that encodes the ray direction and camera-space coordinate. This requires evaluating the MLP for each observation of each point. Our approach, based on an occupancy volume, requires only performing a tri-linear interpolation for each observation, which is significantly cheaper. We train a variant based on this MLP conditioning, replacing the weighting by occupancy score. It increases the memory requirement by 4, since we use 4 observing views per point ($|\mathcal{N}_k|{=}4$). We thus need to halve the batch size and the number of height planes.

We train all variants, including our model, for an identical number of steps. To save compute resources, we train for fewer steps than in the main paper (200k vs 400k) and we use the smaller ResNet-50 architecture with only ground-level inputs. Tab. 2 shows that each of these variants yields a lower positioning accuracy than our model.

**Vertical pooling:** In our design, the vertical pooling is performed with max pooling. Tab. 3-top shows that average pooling is significantly less effective. We hypothesize that averaging makes it harder to ignore features of points located in empty space. We found that pooling with an attention mechanism [85] performs similarly as max pooling despite the increase in computation. Harley

| Variant | Easy (25%) | Med. (50%) | Hard (25%) | All (100%) |
|---|---|---|---|---|
| **SNAP (StreetView-only)** | **39.8 / 51.8** | **36.5 / 47.2** | **22.3 / 32.3** | **34.0 / 44.9** |
| Fixed-height plane (a) | 34.0 / 45.4 | 29.6 / 39.8 | 18.6 / 28.0 | 28.2 / 38.6 |
| No monocular occupancy (b) | 29.8 / 41.7 | 26.1 / 37.3 | 16.1 / 25.7 | 24.7 / 35.8 |
| No multi-view variance (c) | 28.8 / 39.1 | 23.5 / 32.6 | 14.0 / 21.7 | 22.7 / 31.8 |
| Ray conditioning (d) | 11.9 / 22.5 | 8.7 / 17.4 | 4.5 / 10.1 | 8.6 / 17.1 |

Table 2: **Ablation study of the ground-level encoder.** We report the single-image positioning AUC up to thresholds (2.5 m/5°) and (5 m/10°). Variant (a) aggregates the information on an horizontal plane at a fixed height below the camera [1, 86]. This is insufficient for overhanging objects, when ground footprints are occluded, or when the scene is not planar. Variant (b) performs multi-view fusion without monocular priors [106, 110]. This makes it impossible to resolve the depth of objects in the single-image query. Variant (c) further drops the variance term and simply averages feature volumes [34]. This makes it harder to resolve surfaces. Variant (d) replaces the occupancy volume by conditioning each observation feature on the ray and distance using an MLP [85]. This is much more expensive and thus constrains both batch size and scene size, which in turn lowers performance.

| Component | Operator | Easy (25%) | Med. (50%) | Hard (25%) | All (100%) |
|---|---|---|---|---|---|
| Vertical pooling | max | 39.8 / 51.8 | 36.5 / 47.2 | 22.3 / 32.3 | 34.0 / 44.9 |
| (StreetView-only) | average | 32.4 / 43.6 | 29.1 / 39.3 | 17.8 / 27.2 | 27.3 / 37.6 |
| Multi-modal fusion | max | 45.9 / 58.6 | 41.5 / 53.0 | 27.3 / 37.9 | 39.3 / 51.0 |
| (StreetView+aerial) | average | 45.5 / 58.4 | 40.9 / 52.6 | 27.8 / 38.8 | 39.0 / 50.9 |

Table 3: **Ablation study on pooling operators.** Top: In the ground-level encoder, pooling the features vertically with the max operator performs much better than averaging, as measured by single-image positioning AUC. Bottom: Fusing StreetView and aerial neural maps with either max or average pooling performs comparably.

*et al.* [34] flatten the vertical elements with a space-to-depth (or pixel-shuffling) operation, followed by an MLP. This makes the model sensitive to a translation along the vertical axis, which rarely occurs in small driving datasets based on a few cars with identical specifications, but matters for heterogeneous data captured by backpacks and cars with widely different setups. We thus found that this approach yields a lower performance than a simpler pooling. This also makes it impossible to adjust the number of height planes at inference time.

**Multi-modal fusion:** Tab. 3-bottom shows that the choice of pooling operator makes little difference when fusing neural maps inferred from StreetView and aerial inputs.

# B Qualitative examples

**Visual positioning:** Due to space constraints, we showcase only a handful of visual positioning examples in the main paper. We show additional visualizations of successfully localized queries in Figs. 8 to 10 and failure cases in Figs. 17 and 18. We also show some examples with non-linear mapping trajectories in Fig. 19, which our method is robust to. As in Fig. 3, we show the 3-DoF poses of map images **(red)**, the GT query pose **(black)**, and the pose predicted by SNAP **(green)** with its error $(\Delta t, \Delta R)$. We visualize neural maps by projecting them to RGB using PCA.

**Sequence to sequence alignment** In the paper we focus on the alignment of single-image queries to maps built from multiple views, but SNAP can arbitrarily align any pair of neural maps. We show two examples of sequence to sequence alignment in Figs. 11 and 12. These maps are built from five views. For ease of understanding, we lift our neural maps into 3D using LiDAR, color-coding each LiDAR point with the neural map values of the cell it belongs to – note that this is stricly for visualization purposes, and our algorithm does not rely on LiDAR.

**Semantic mapping:** We show additional visualizations of the semantic maps predicted by our model after fine-tuning it in only 2k scenes in Figs. 14 and 15.

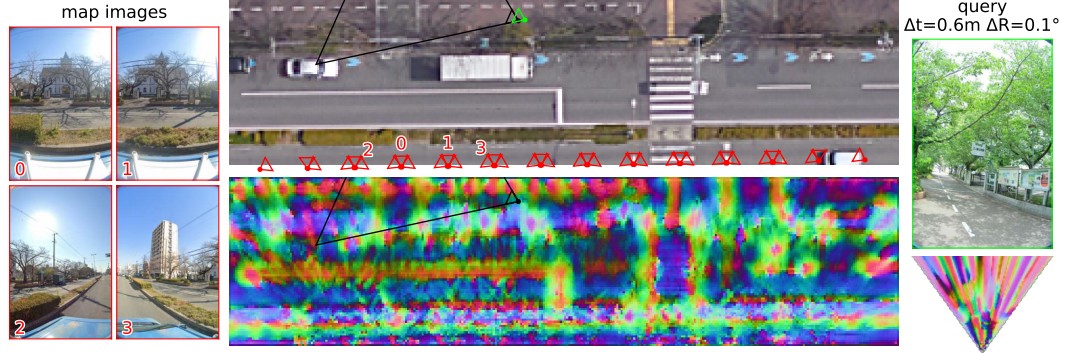

a) The query is localized within 0.6 m from a different road (medium).

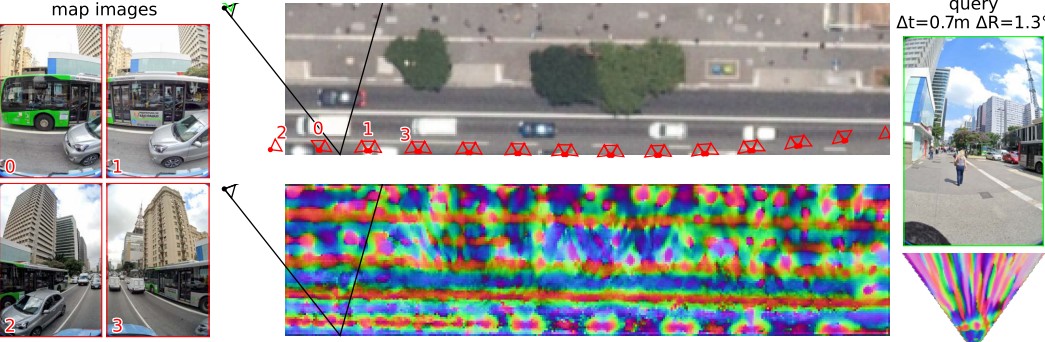

b) The query is localized within 0.7 m despite significant occlusions in the reference views (medium).

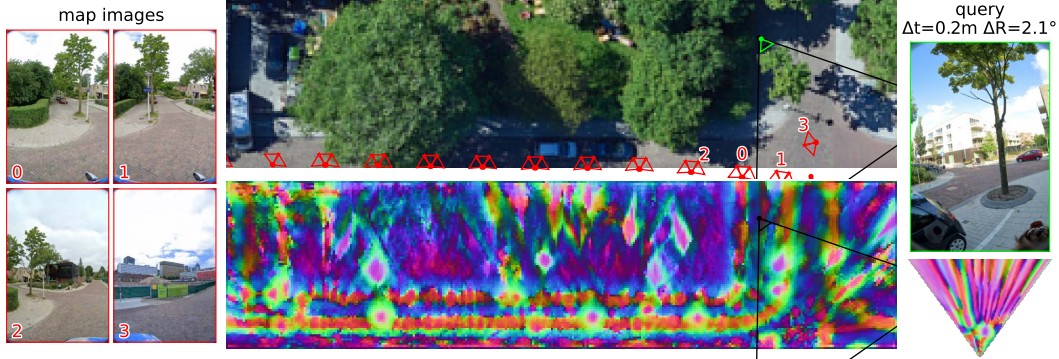

c) The tree is clearly visible in both the query and the reference neural maps (hard).

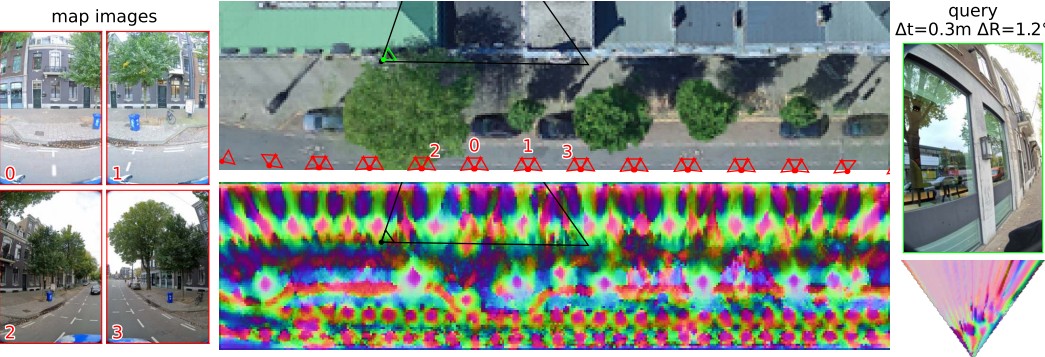

d) The query is correctly localized despite drastic perspective changes and reflections in the window (medium).

Figure 8: **Single-image localization (1/3).** We show successful examples. See legend in Fig. 3.

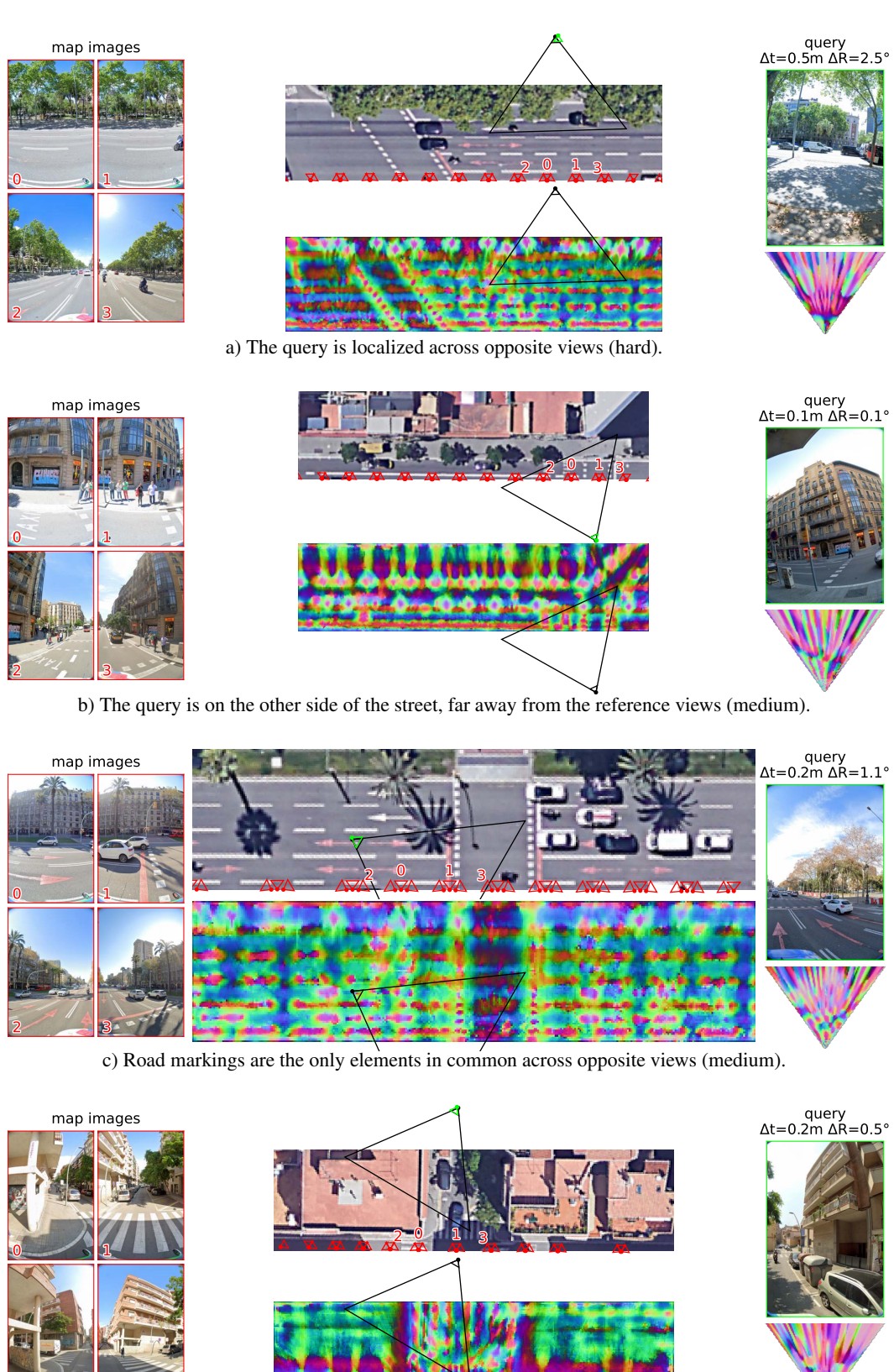

a) The query is localized across opposite views (hard).

b) The query is on the other side of the street, far away from the reference views (medium).

c) Road markings are the only elements in common across opposite views (medium).

d) The query is correctly localized across a different street and despite drastic perspective changes (hard).

Figure 9: **Single-image localization (2/3).** We show successful examples. See legend in Fig. 3.

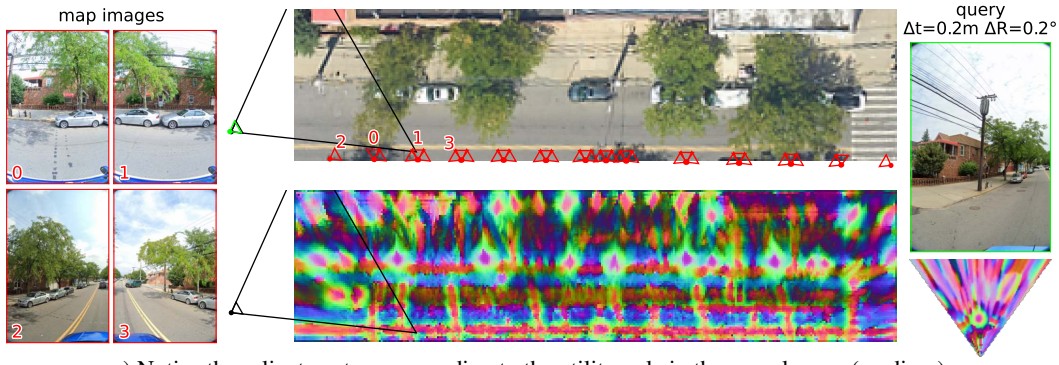

a) Notice the salient spot corresponding to the utility pole in the neural maps (medium).

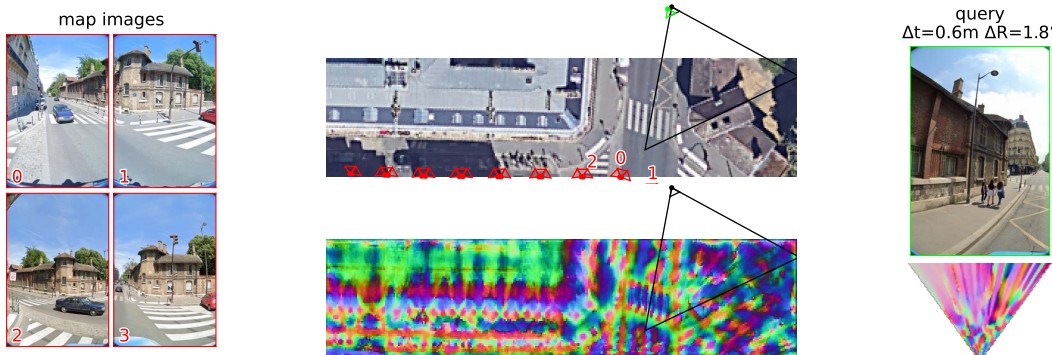

b) Notice that the building facade closest to the query is not visible from the reference views (hard).

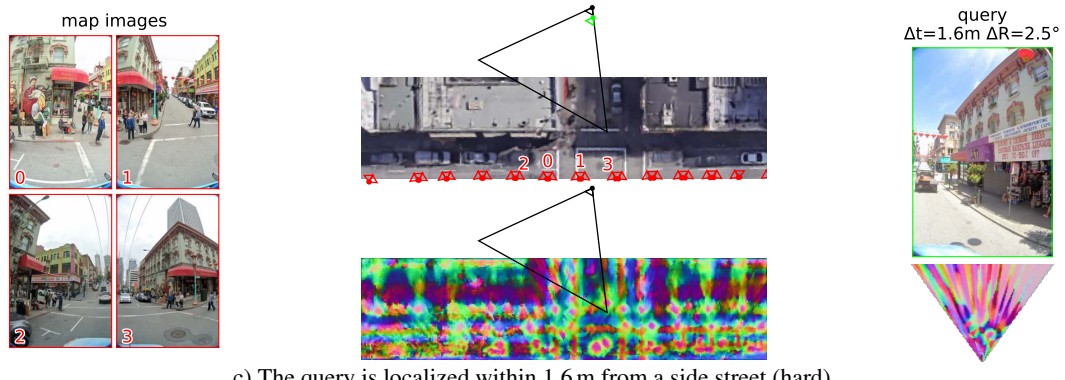

c) The query is localized within 1.6 m from a side street (hard).

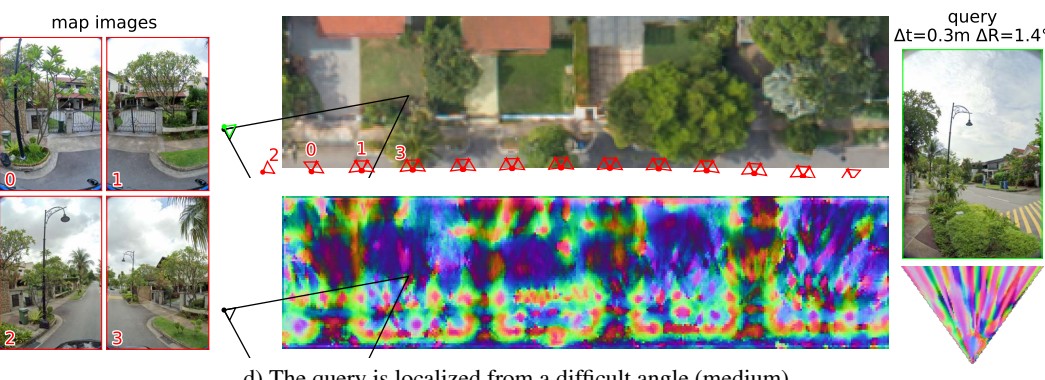

d) The query is localized from a difficult angle (medium).

Figure 10: **Single-image localization (3/3).** We show successful examples. See legend in Fig. 3.

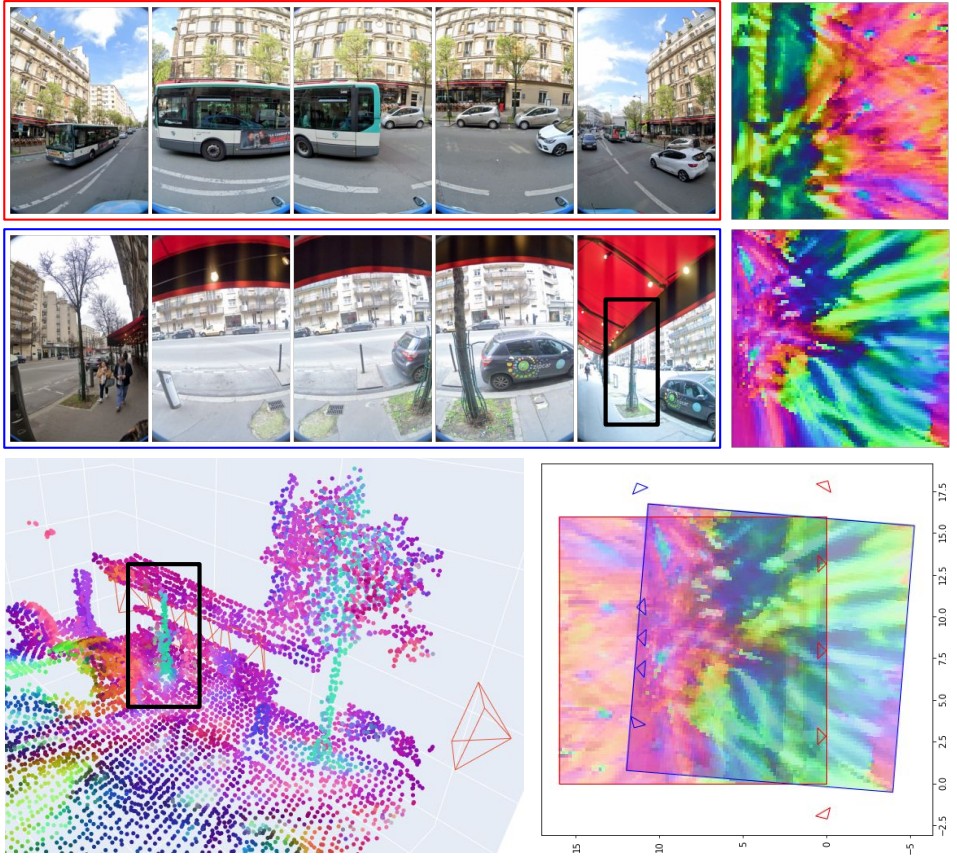

Alignment error: $\Delta t$=0.5m, $\Delta R$=0.33$^o$

Figure 11: **Sequence-to-sequence alignment (1/2).** We align a car sequence (red) to a backpack sequence (blue) across opposite views. Notice how the neural maps (top right) are clearly similar despite the large viewpoint difference, and can be easily aligned (bottom right). In the bottom left we lift the neural map for the first sequence into 3D by coloring a LiDAR point cloud with the RGB value of the grid cell it belongs to – note that this is strictly for visualization purposes, and our method *does not* rely on LiDAR. Notice how the semantics learned by our model correspond to real scene features, such as the pole (in cyan, enclosed in a black box).

**Large-scale mapping:** We can easily build large tiles by 'stitching' smaller neural maps together via cell-wise max-pooling, similarly to how we fuse aerial and ground-level neural maps. Starting from multiple sequences posed in a common reference frame, Fig. 13 shows how neural maps are inferened for each of them and finally combined together.

## C   Detailed results

Fig. 21 shows the single-image positioning recall performance for each for the 6 test cities. The performance of the different approaches is fairly consistent across cities, with the exception of Osaka. We observe that Osaka has a large number of small, narrow streets, and StreetView sequences often feature close-ups of building facades: see Fig. 18-(c-d). This is somewhat out of distribution with the training set and could be solved by training on more data. It also explains why structured-based matching approaches are either very accurate (when there is sufficient visual overlap between query and reference views) or fail to register the image (when there isn't).

Fig. 22 shows the recall for queries taken by cameras mounted on either backpacks or cars. Backpack queries have a lower recall because they are typically taken from a sidewalk and thus have larger viewpoint differences (Fig. 16).

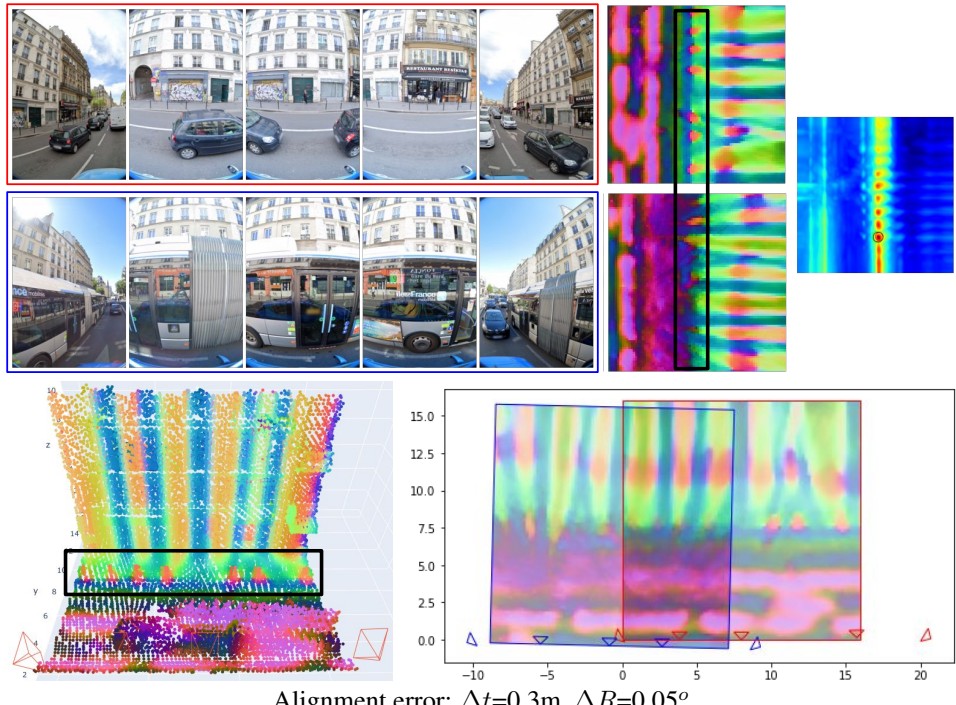

Alignment error: $\Delta t$=0.3m, $\Delta R$=0.05$^o$

Figure 12: **Sequence-to-sequence alignment (2/2).** We align two car scenes, the second of which is heavily occluded due to a passing bus. Notice how the neural maps (top right) are visually similar and can be easily aligned. The neural map for the first sequence clearly shows a row of poles, which are occluded in the second sequence (black box). Our method is robust and can align the sequences using road markings and building boundaries. On the top right we plot an alignment heatmap by running exhaustive aligment in 3-DoF between both neural maps (we max-pool over rotations, for ease of understanding): notice the clear maximum (circled), with smaller maxima along the road. In the bottom left we lift the neural map for the first sequence into 3D by coloring a LiDAR point cloud with the RGB value of the grid cell it belongs to – note that this is strictly for visualization purposes, and our method *does not* rely on LiDAR.

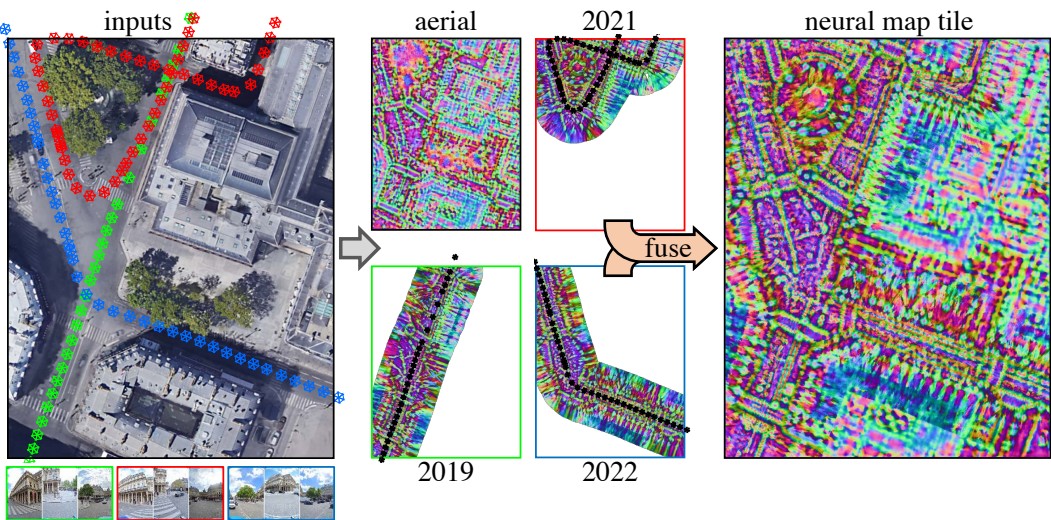

Figure 13: **Building neural tiles.** We combine an aerial view and car sequences captured over multiple years into a single neural map that spans a large area. An arbitrary number of inputs can be combined in such way.

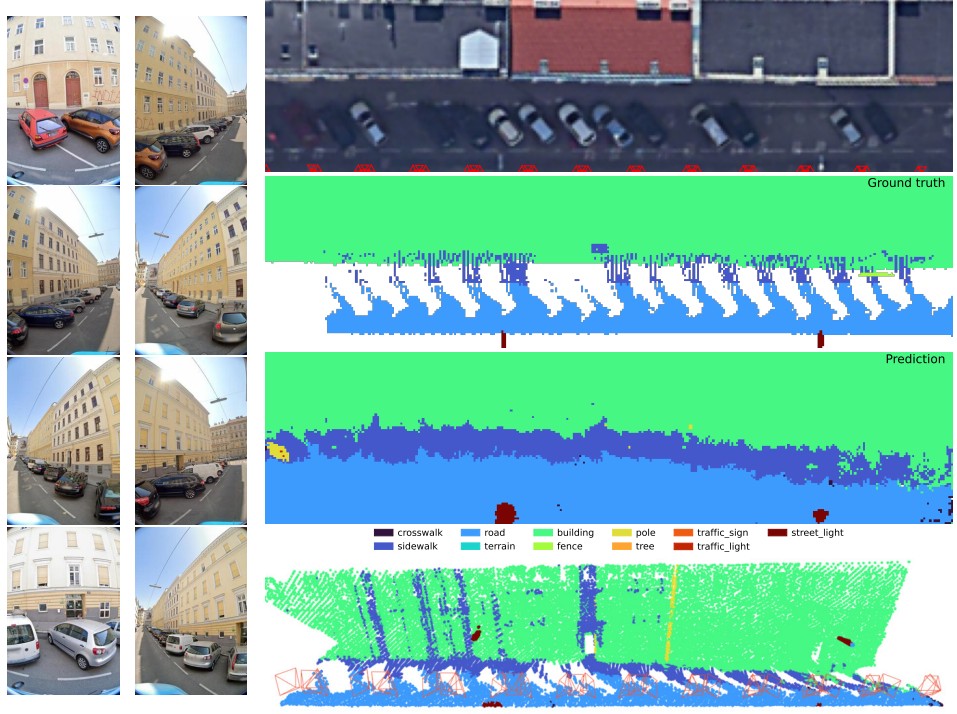

a) The curb is mostly occluded by the parked cars and has very sparse ground truth labels. Notice how our neural maps encode the location of street lights that are hanging above the street.

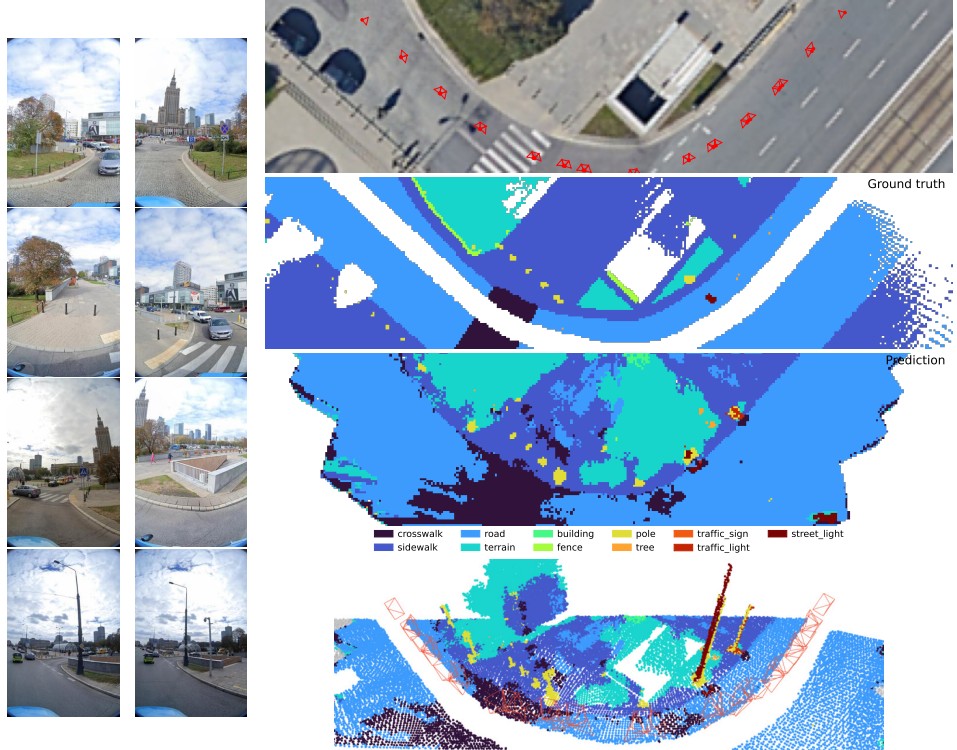

b) Because of the bike lane and confusing markings at the intersection, the model incorrectly segments the crosswalk.

Figure 14: **Semantic mapping (1/2).** 2D segmentation predicted from pre-trained neural maps by a small CNN trained in a supervised fashion with GT semantic labels of 2k scenes. We also show 3D lidar point clouds colored by the predicted segmentation (lidar is not used as input).

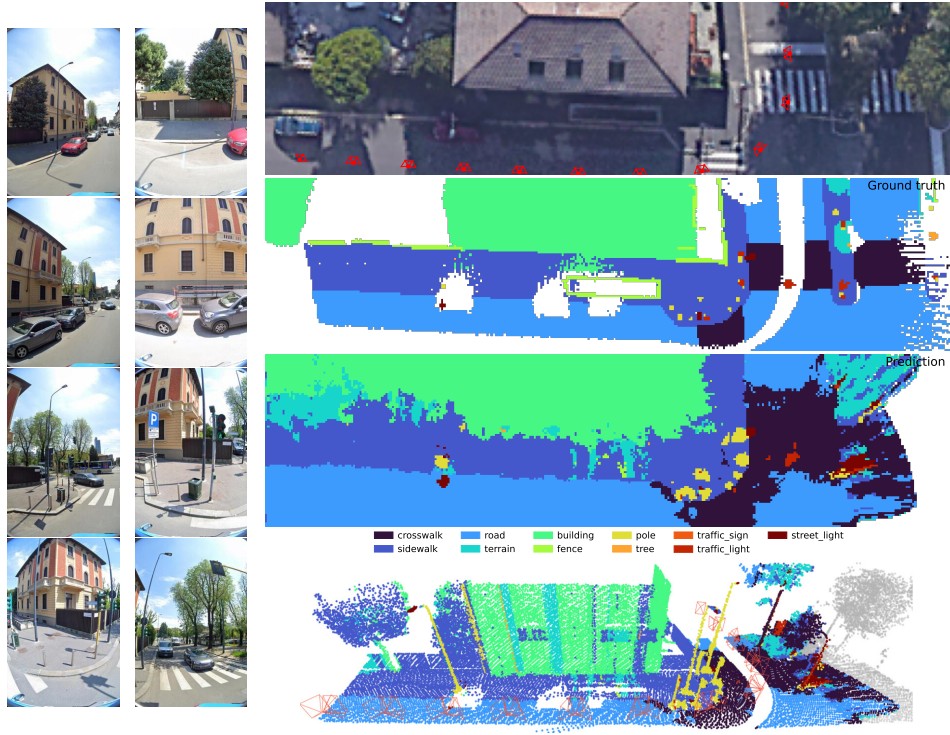

a) The StreetView car turns around the corner, so the right side of the map is behind the cameras and thus very sparsely observed. Areas with good visual coverage are much better segmented.

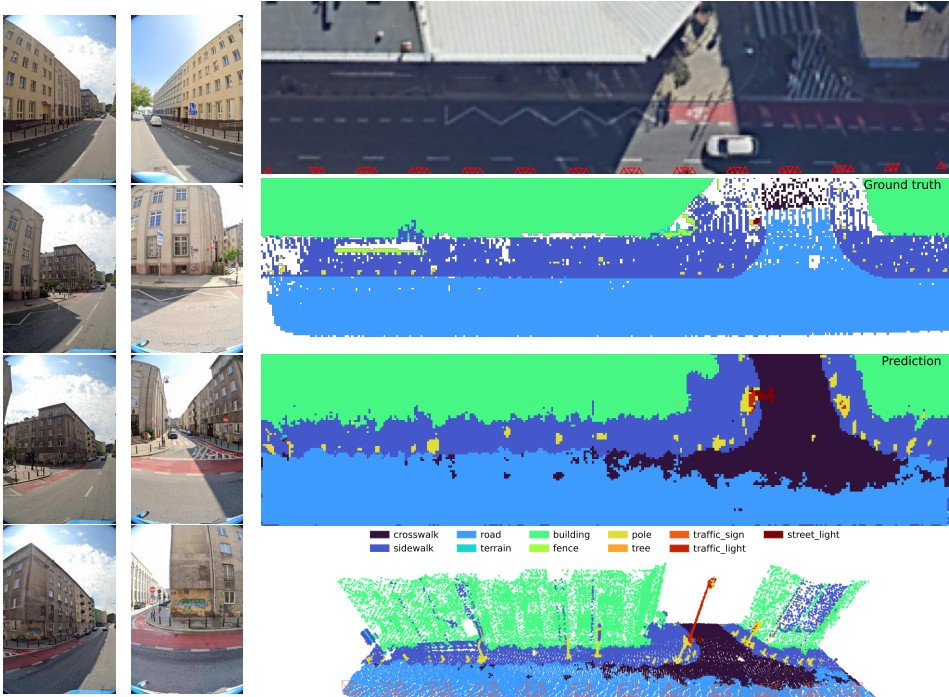

b) The model roughly segments road, sidewalk and buildings, and correctly resolves small objects such as the poles or the overhanging traffic light. It gets confused around the sidewalk, which has multiple striped patterns that are also difficult to interpret by the human eye.

Figure 15: **Semantic mapping (2/2).** 2D segmentation predicted from pre-trained neural maps by a small CNN trained in a supervised fashion with GT semantic labels of 2k scenes. We also show 3D lidar point clouds colored by the predicted segmentation (lidar is not used as input).

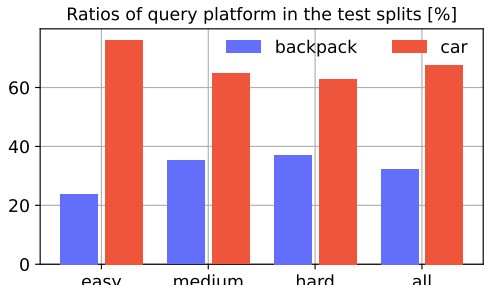
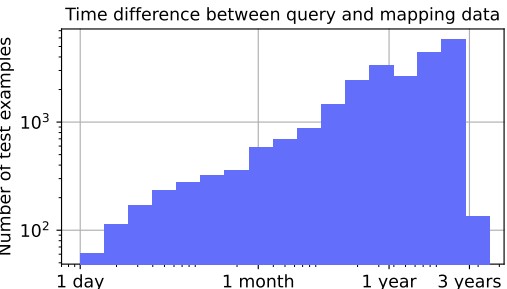

Figure 16: **Large viewpoints and temporal differences.** Left: The easy split contains more car-mounted queries, and the hard split contains more backpack queries. This make sense, as examples captured from backpacks are typically captured from sidewalks instead of the road, which results in difficult localization scenarios across opposite views. Right: The time difference between query and mapping images spans from a few days to a few years. This yields challenging localization scenarios with large appearance and even structural changes, *e.g.*, due to construction work.

# D    Dataset

We provide more details about the data used for the experiments conducted in the paper.

**Platform type:** Ground-level mapping sequences are selected from sequences captured by cars only. Query images are sampled from sequences captured by either cars or backpacks. For each map tile, we sample as many car queries as backpack queries, when possible. As cities are only partially covered by backpack sequences, the final ratio of backpack queries varies per city, from 20% to 40%. Fig. 16-left shows the ratio of both types for the different difficulty splits of the test set.

**Map grid:** We found that maps of size $64 \times 16$ m provide a good trade-off between visual diversity and memory consumption. Since multi-view frames are provided every ~5 m, segments of 14 frames generally cover a length of 64 m. We therefore sample segments of 14 frames and define the map grid to be aligned with the average pose of the 7[th] and 8[th] side cameras. Given that most car segments are rectilinear, the map is most often aligned with the mapping views. In turns, the coverage of the map varies and can make the localization more challenging, which is generally a good property for both training and evaluation. Fig. 19 shows examples of maps with partial coverage.

**Spatial distribution:** For each of 11 training cities, we define disjoint training and validation regions. We show these regions in Fig. 20. We sample at most 5M training examples from each training region. The validation region corresponds to 5 (10 for New York) randomly-sampled level-14 S2 cells[§], from which we sample 1024 examples. 4k test queries are sampled from 20 random S2 cells in each of the 6 test cities, as shown in Fig. 23. This corresponds to an average area of $5.6 \text{ km}^2$ per city.

**Temporal distribution:** Sequences used for mapping or localization are randomly drawn from the pool of available StreetView data. Fig. 16-right shows the distribution of temporal differences. This yields challenging queries with large appearance changes, *e.g.*, due to the seasons, or structural changes, *e.g.*, due to construction work. Aerial images are selected as captured as close as possible to the ground-level mapping images. The average time difference varies from 6 months to 3 years.

**Ground truth semantic rasters:** In the semantic mapping experiments, we used a small labeled dataset to train and evaluate our fine-tuning. We start with 1.5k segments of data captured in 84 cities across the world by StreetView cars equipped with LiDARs. We aggregate the measurements into point clouds that we label into 10 classes spanning objects encountered in urban environments. These classes include surfaces with a direct ground footprint (road, sidewalk, crosswalk, building, tree) and small overhanging objects (fence, pole, street light, traffic sign, traffic light). As each segment is about 80 m long, we derived two maps of size $64 \times 16$ m from each, facing either side of the street. We derived semantic rasters by binning the points into each map cell and inferring two types of labels: a) Class surfaces are mutually-exclusive in 2D so we compute multi-class surface labels corresponding to the class most represented in each map cell. b) Differently, objects might appear on top of each other so they are not mutually-exclusive. We thus compute binary labels for each object class using a threshold on the number of points in each map cell. We obtain 3k semantic rasters which are split between training and test sets, each including data from all 84 cities.

---

[§]As defined by Google's S2 geometry library: `https://s2geometry.io`

**Semantic rasters for OrienterNet:** To obtain labels at the scale required to train OrienterNet [78], we apply this rasterization process to sparse Structure-from-Motion points that have similar semantic labels. Those labels are obtained fully automatically by fusing 2D image segmentations, without any human work [30]. While the resulting rasters are noisier than those derived from LiDAR, they cover our entire training set rather than only 3k maps. They also more consistently cover small objects (like trees and street lights) than OpenStreetMap.

# E   Implementation details

**Training:** We develop our models with JAX [11] and Scenic [22], and format our dataset with TFDS [97]. All models are trained on 16 A100 GPUs over 3-4 days with a total batch size of 32 (2 examples per GPU). We use the ADAM [44] optimizer over 400k iterations for the small model and 200k iterations for the larger one. This corresponds to 24% and 12% of the training data, respectively, but is sufficient for convergence. We use a constant learning rate of $10^{-4}$ for the first half and then decay it to zero with a cosine decay. Streetview images are resized to $684 \times 456$. We train our models with mixed precision and use gradient checkpointing in order to reduce GPU memory usage as much as possible. The temperature $\tau$ is parameterized as $\log \tau$, initialized to 0, and optimized alongside other model parameters. All hyperparameters are tuned on the validation set.

In the multi-view fusion, we fuse observations from the 4 closest views that observe a point. This saves a significant amount of memory compared to fusing observations from all views. Using fewer views (20 instead of 36 at inference time) also makes the problem artificially harder.

**Pose estimation:** We use 10k RANSAC samples during training and 20k during inference. Since only 3 constraints are required to solve for a 3-DoF pose, 2 2D correspondences over-constrain the problem. Randomly sampling each correspondence independently thus yields a large number of invalid poses. To increase the ratio of valid poses, we sample $8\times$ the number of desired correspondences and pick the ones with the ratio of distances closest to 1. This improves the pose quality, at both training and test time, and is well amenable to parallelized GPU processing, without requiring an expensive sorting. When computing the featuremetric score, query points that project outside the map are given the value of the map boundary. We found this to be more robust than ignoring these points. At inference time, we refine the pose by exhaustive search in a grid centered at the maximum-score pose, with size $4\,\mathrm{m}\times4\,\mathrm{m}\times5°$ and resolution $20\,\mathrm{cm}\times0.5°$.

**Architecture:** The decoder of the image encoder is an FPN with bilinear upsampling. We progressively upsample features from stages 4, 3, and 2 to the resolution of stage-1 ResNet features, which corresponds to an output stride of 4. For the aerial encoder, we remove the root block ($7\times7$ convolution and max pooling) such that the output stride is 1.

**Semantic fine-tuning:** We train a small 2-layer CNN with two heads: one for the multi-class classification of ground surfaces, using a softmax with cross-entropy loss, and the other for the multi-label classification of objects, using sigmoids and binary cross-entropy losses. We train for 5k iterations and retain the checkpoint with the lowest validation loss.

# F   Social impact

**CO2 emissions:** Experiments were conducted using Google Cloud in region us-central1, which has a carbon efficiency of 0.57 $kgCO_2eq$/kWh. A cumulative of 88.8k hours of computation was performed on NVidia A100 GPUs. Total emissions are estimated to be 12654 $kgCO_2eq$, of which 100 percent were directly offset by Google. Had this model been run in region europe-west6, the emissions would have been of 355.31 $kgCO_2eq$. Estimations were conducted using the ML Impact calculator [47].

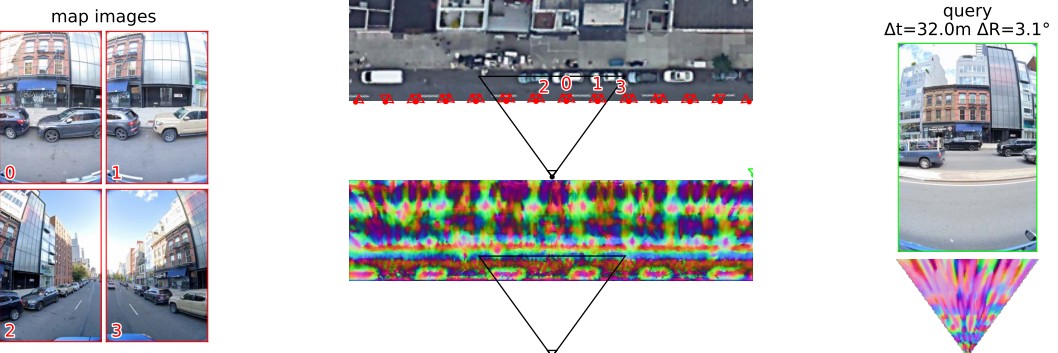

a) The query is too far from the buildings and fails to localize using features along the road (medium).

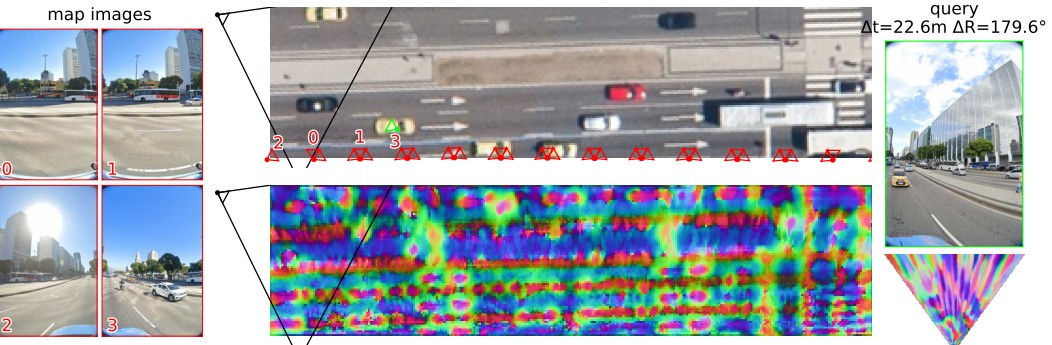

b) SNAP match the right visual content from the wrong side, due to scene symmetries (medium).

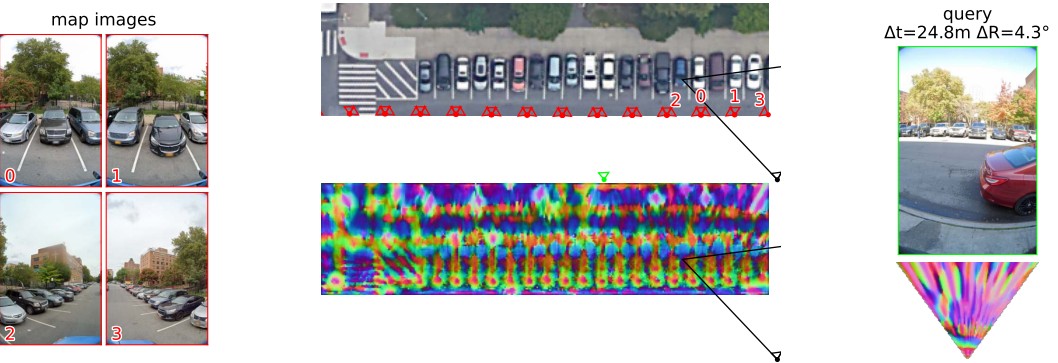

c) Reasonable but incorrect guess along a row of parking spots with very little structure otherwise (medium).

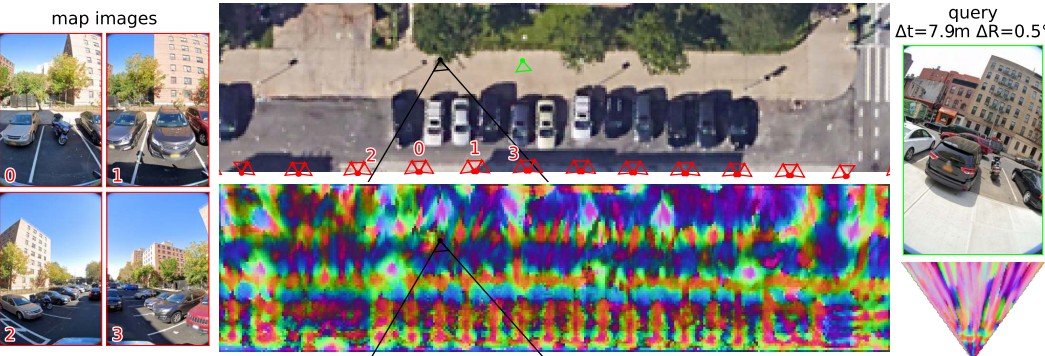

d) Similar to (c), but across opposite views, and thus more challenging. Note that the rotation error is <1° (hard).

Figure 17: **Localization failures (1/2).** See legend in Fig. 3.

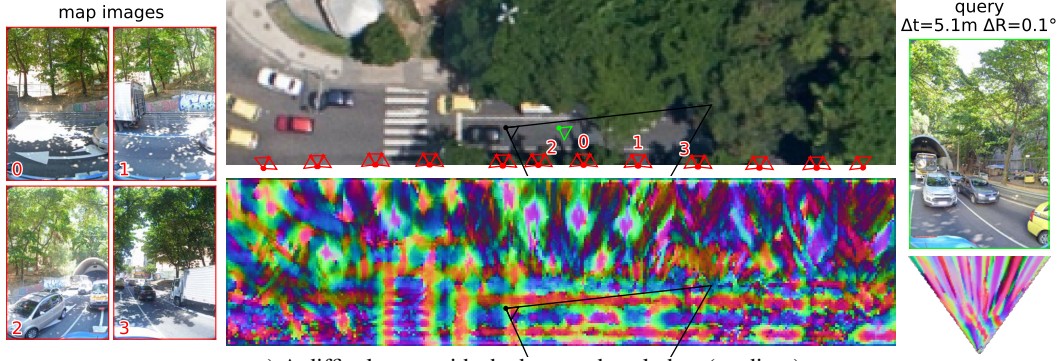

a) A difficult case with shadows and occluders (medium).

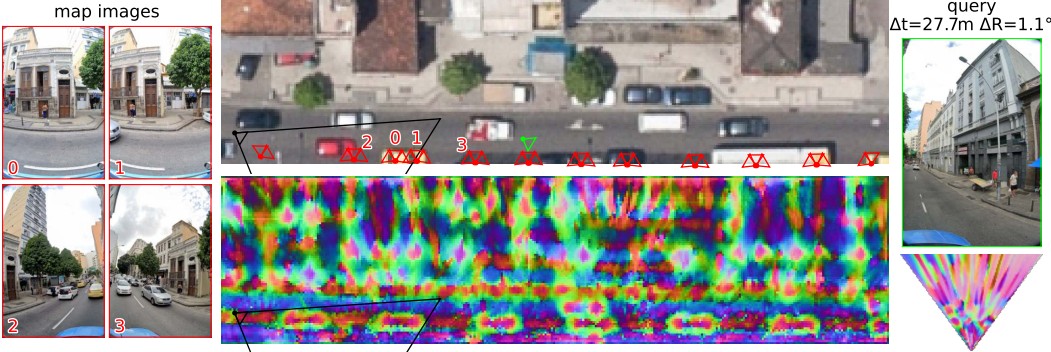

b) Correct orientation (1°) at the wrong location along the road, across opposite query/reference views (medium).

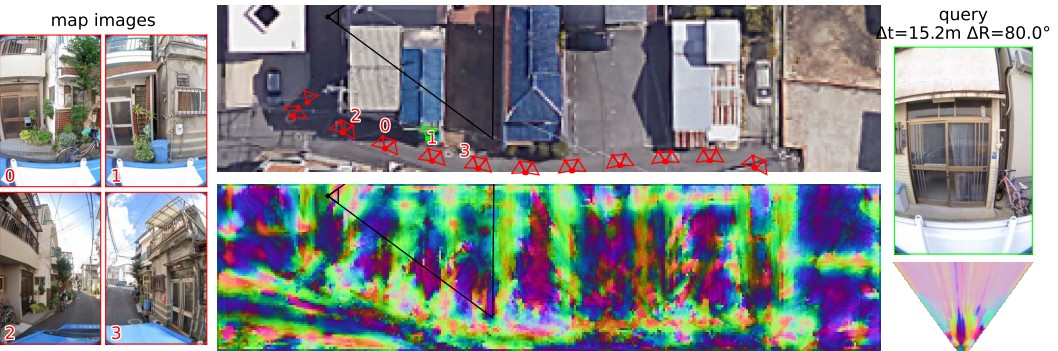

c) [Osaka] Query and reference views are very close to the buildings, which is rarely seen during training (easy).

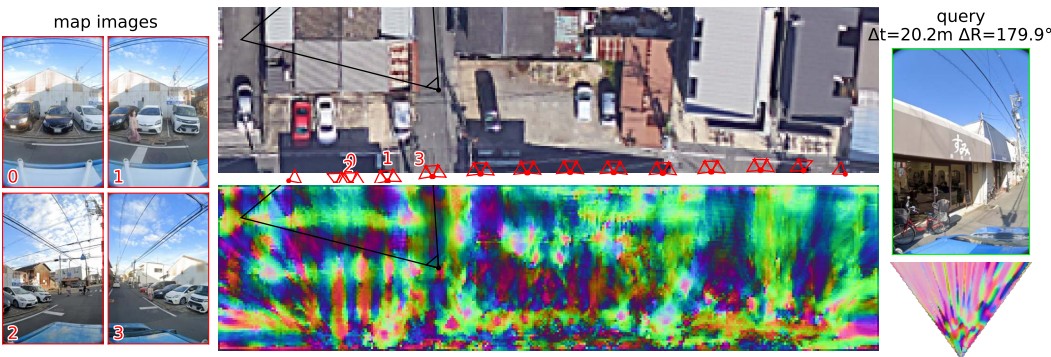

d) [Osaka] Query is on a side street with little visual overlap with the reference views (easy).

Figure 18: **Localization failures (2/2).** See legend in Fig. 3.

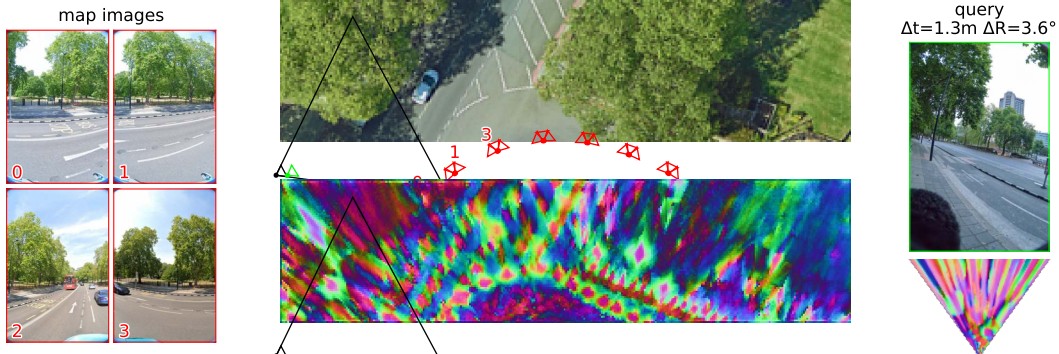

a) The tree is smeared yet clearly visible on the *left* side of the query BEV as a pink dash in a green circle (hard).

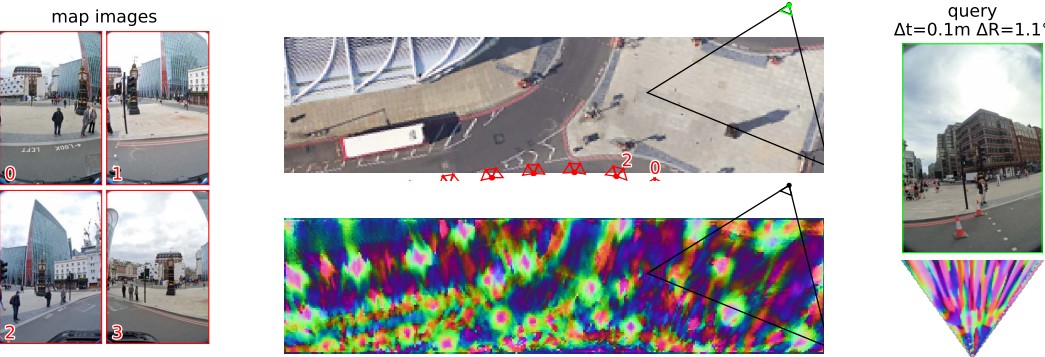

b) The query is accurately localized (10 cm) from the other side of the square (hard).

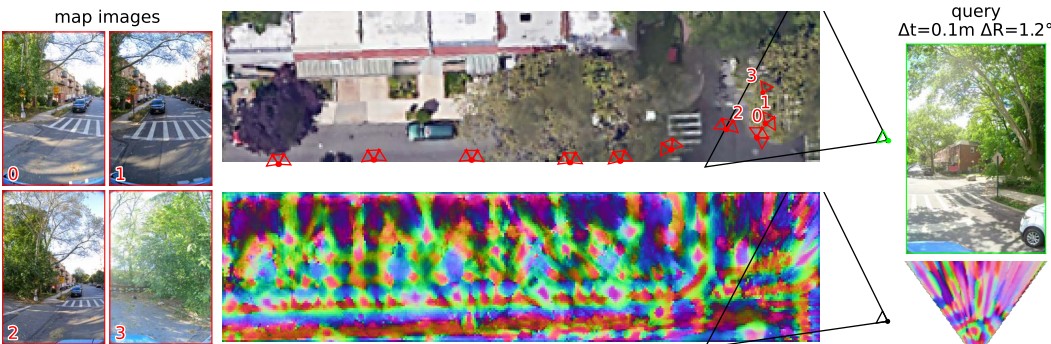

c) The reference views are irregularly distributed due to the car's motion, to which our method is robust (hard).

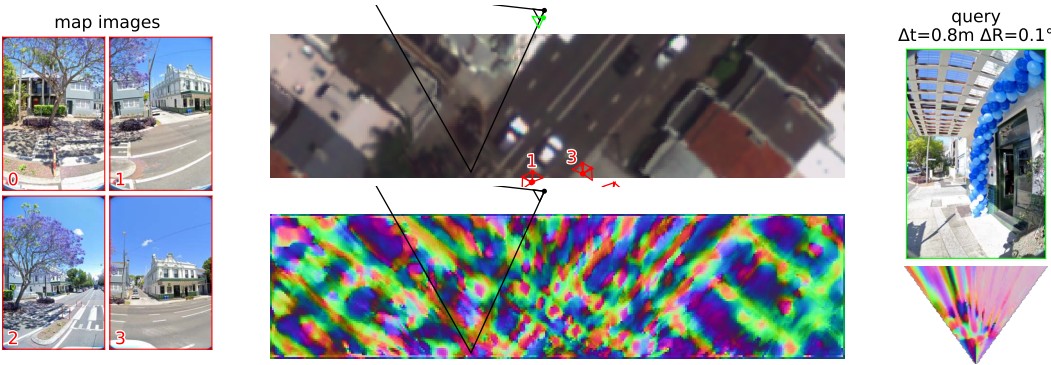

d) The query is localized within 0.8 m from a very difficult angle with little visual overlap (hard).

Figure 19: **Single-image localization.** Examples of scenes with non-rectilinear mapping sequences, which results in partial coverage. Please refer to Appendix D for details.

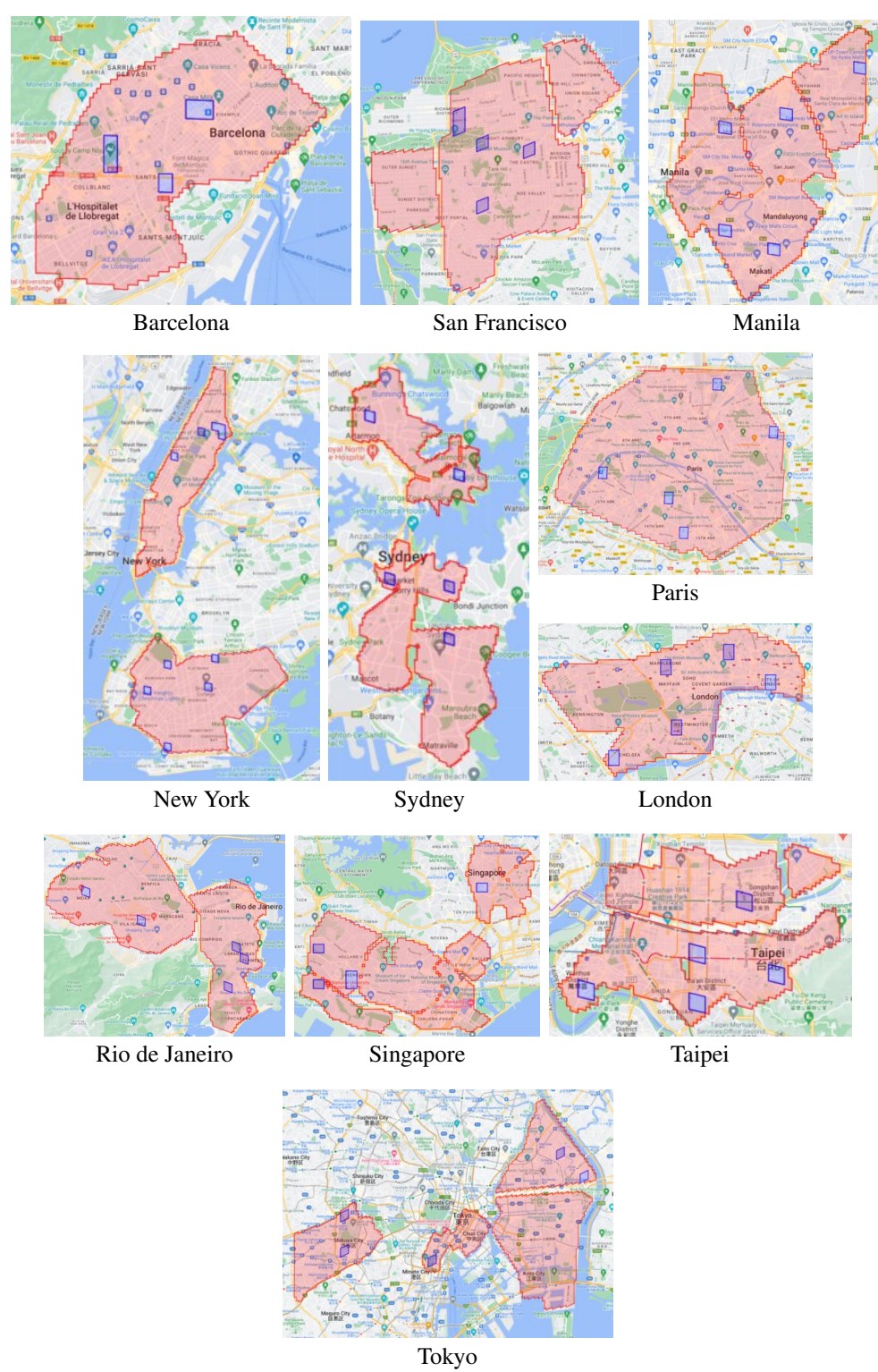

Figure 20: **Spatial distribution of the training data.** In each city, training examples are sampled from the red areas while validation examples are sampled from the blue areas.

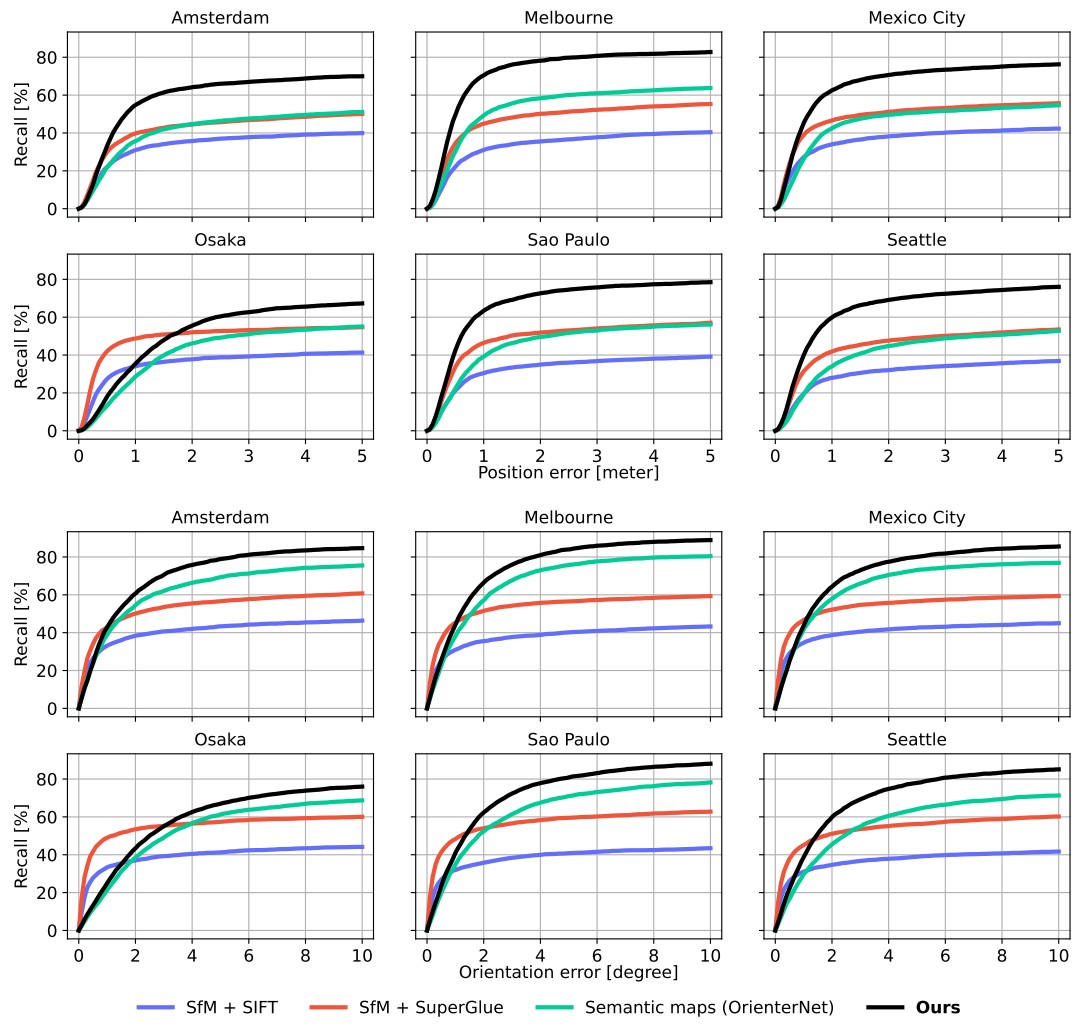

Figure 21: **Single-image positioning per city.** We plot the position and orientation recall for each of the 6 test cities.

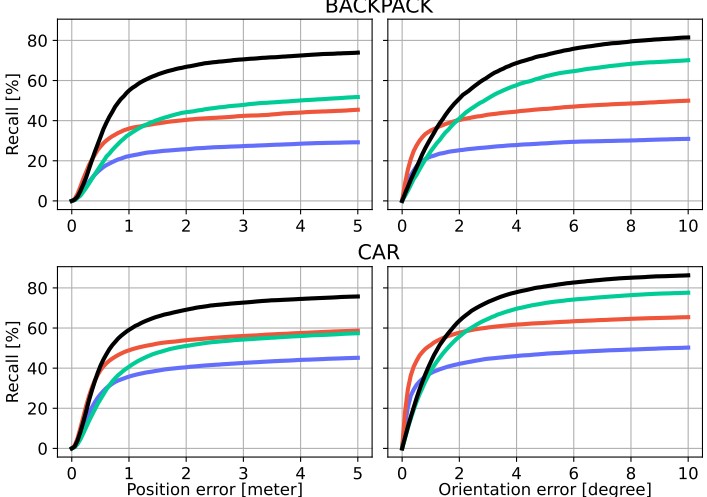

Figure 22: **Single-image positioning per platform.** We plot the position and orientation recall for queries taken by cameras mounted on either backpacks or cars.

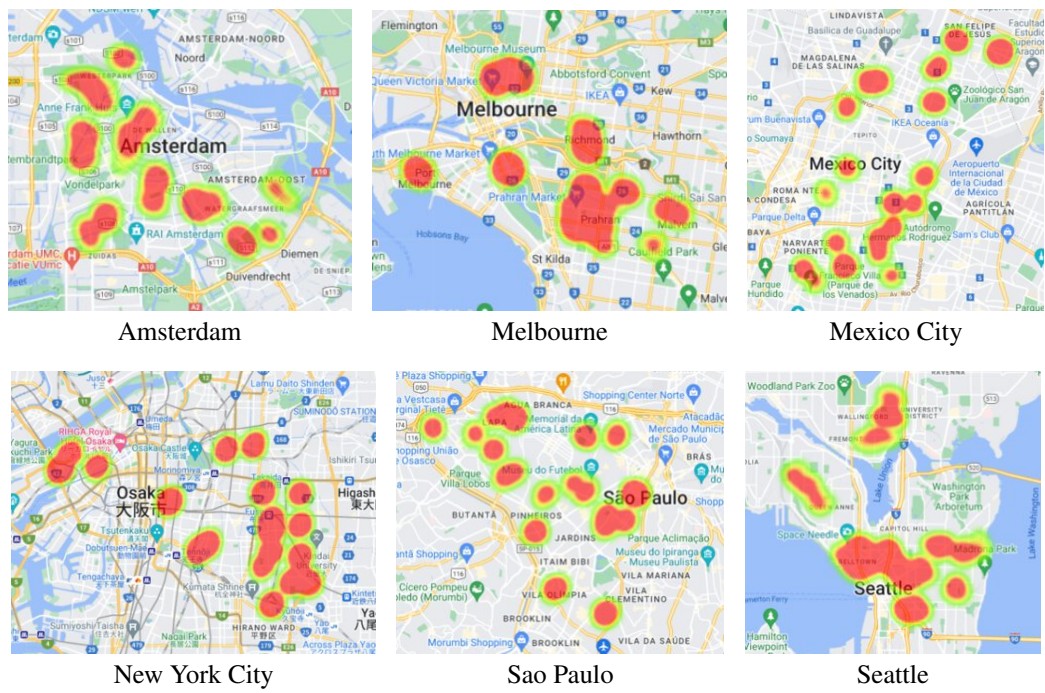

Figure 23: **Spatial distribution of the test data.** For each city, test queries are sampled from 20 level-14 S2 cells and are here shown as heatmaps.

