# OpenReview forum: "SNAP: Self-Supervised Neural Maps for Visual Positioning and Semantic Understanding"
_NeurIPS.cc/2023/Conference — NeurIPS 2023 poster_

### Official Review · Reviewer_PKpW · 2023-07-02

**Soundness:** 2 fair
**Presentation:** 2 fair
**Contribution:** 2 fair
**Rating:** 4
**Confidence:** 4

**Summary:**

 Currently，2D semantic maps have limitations:  they lack detail, often contain inaccuracies, and are difficult to create and maintain， in order to solve these problem,  the author proposes fusing features from different input modalities to obtain a 2D neural map that is robust over time and retains sufficient geometric and semantic information. The effectiveness of the method was demonstrated by using the 2D neural map generated by SNAP for other tasks.

**Strengths:**

  This paper proposes to fuse various input modalities and use camera pose estimation to generate a 2D neural map, which is the originality. Moreover, the paper delves into this methodology, related work, and conducts a plethora of experiments, culminating in high-quality output. However, the clarity of the paper requires further refinement, and its significance remains somewhat limited.

**Weaknesses:**

The author demonstrated the effectiveness of using self-supervised pre-training in the article, but there is no relevant experimental evidence to prove whether the fusion of multi-mode features and the proposed supervision method are reasonable. In addition, it is unfair to only compare with OrienterNet in Experiment 1, as the generation of neural graphs involves aerial images, which are more similar to [1][2].
[1] Shi Y, Li H. Beyond cross-view image retrieval: Highly accurate vehicle localization using satellite image[C]//Proceedings of the IEEE/CVF Conference on Computer Vision and Pattern Recognition. 2022: 17010-17020.
[2] Lentsch T, Xia Z, Caesar H, et al. SliceMatch: Geometry-guided Aggregation for Cross-View Pose Estimation[C]//Proceedings of the IEEE/CVF Conference on Computer Vision and Pattern Recognition. 2023: 17225-17234.




**Questions:**

1.It is necessary to supplement experimental evidence for the effectiveness of fuse and contrastive learning supervision in the generation process of neural maps.
2.Can other satellite-based localization methods generate neural maps? The authors need to conduct experiments to demonstrate the advantages of their method compared to other satellite-based localization methods in generating neural maps.
3.Does the method also perform well on other general datasets such as KITTI?


**Limitations:**

The author discussed the limitations of the method on accuracy and the impact of the method on privacy.

---

> ### Author Rebuttal · Authors · 2023-08-09
>
> **[...] there is no relevant experimental evidence to prove whether the fusion of multi-mode features and the proposed supervision method are reasonable.
> It is necessary to supplement experimental evidence for the effectiveness of fuse and contrastive learning supervision in the generation process of neural maps.**
>
> We show the effectiveness of multi-modal fusion _precisely_ with experimental evidence. Table 1 shows that fusing ground-level and aerial cues improves performance by ~8% relative to using only ground-level cues. Likewise, we show that our supervision method is reasonable through our experiments: we outperform traditional SfM-based methods in very challenging situations (and overall) by training only with posed imagery within the contrastive learning framework we present. It is not clear to us how one would alternatively learn neural maps given only image poses.
>
> The ablation study in Section A of the supplementary material demonstrates the effectiveness of our design decisions. We invite the reviewer to clarify what they mean with these statements and we will try to address them during the discussion phase.
>
> **In addition, it is unfair to only compare with OrienterNet in Experiment 1, as the generation of neural graphs involves aerial images.**
>
> We believe our comparison with OrienterNet is fair. Please note that OrienterNet takes as input 2D, overhead semantic maps (the kind used for navigation), _not_ aerial images. We train our version of OrienterNet with semantic rasters similar to the semantic maps from OpenStreetMap used in their paper. While our rasters are noisy they provide a consistent, global coverage of fine-grained classes relevant for localization (e.g. tree, streetlight, poles). As stated in L253, we checked our implementation with the authors.
>
> **Can other satellite-based localization methods generate neural maps? The authors need to conduct experiments to demonstrate the advantages of their method compared to other satellite-based localization methods in generating neural maps.**
> * **[1] Beyond cross-view image retrieval: Highly accurate vehicle localization using satellite image, Shi et al, CVPR 2022**
> * **[2] SliceMatch: Geometry-guided Aggregation for Cross-View Pose Estimation, Lentsch et al, CVPR 2023**
>
> Table 1 reports the performance of a variant of SNAP trained and evaluated using only aerial images. This model is conceptually similar to the approach of Fervers et al [21] (aerial/satellite map, BEV-to-map matching, with known gravity), which itself yields significantly more accurate localization than the approaches of both Shi et al. and Lentsch et al on identical or similar datasets (FordAV, KITTI 360). We thus believe that evaluating these works is unnecessary. Conceptually, both approaches have a limited performance: Shi et al make restrictive assumptions that do not hold in practice (strong location and orientation prior) while SliceMatch lacks geometric inductive bias to produce fine-grained predictions (no explicit BEV matching).
>
> Moreover, our experiments show that using only ground-level imagery performs ~46% (relative) better than using only aerial imagery, and that combining them improves the performance over ground-level only by a more modest ~8% (relative). This is due to the loss of information from occluders (particularly vegetation) in overhead images, which are inevitable in cluttered urban environments. This is a large gap and, we believe, a difficult one to bridge with better architectures that still rely _only_ on aerial/satellite imagery.
>
> Finally, please note that SliceMatch [2] is a CVPR 2023 paper. Code was not available until June 2023, whereas the NeurIPS submission deadline was in May 2023 (please refer to their GitHub repository).
>
> **Does the method also perform well on other general datasets such as KITTI?**
>
> While we plan to tackle this in the future, the domain gap is significant and this is out of the scope of what we can achieve within the rebuttal window.

---

> > ### Comment · Reviewer_PKpW · 2023-08-18
> > **Reviewer response to rebuttal**
> >
> >    Thank you for answering my concerns. Please find my following comments.
> >
> > **1. Effectiveness of multimodality.**
> > In my opinion, the author uses more modalities as input and wishes to demonstrate that the proposed method is superior to any single-modality approach. In Figure 1 of the paper, the author mentions the combination of street view and aerial images. However, in the experimental data presented in Table 1, the author demonstrates that their generated semantic maps are better than using the street view method and also better than using the semantic method, but they do not compare other methods with using only aerial images. Although the author responses that such a comparison is not necessary, I would prefer to see some quantitative results.
> >
> > **2. Unfair comparison.**
> > In the visualization results of Figure 4, the representation of pillars and trees on the map is very clear, which is crucial for successful localization and image segmentation. I believe this is mainly due to the utilization of aerial images and street views. On the other hand, OrienterNet only uses semantic maps as input, making it almost impossible to learn the positions of pillars and trees in such maps. Therefore, I think the author should pay more attention to comparing methods that employ aerial images and street views as inputs.

---

> > > ### Author Response · Authors · 2023-08-18
> > >
> > > **However, in the experimental data presented in Table 1, the author demonstrates that their generated semantic maps are better than using the street view method and also better than using the semantic method, but they do not compare other methods with using only aerial images. Although the author responses that such a comparison is not necessary, I would prefer to see some quantitative results.**
> > >
> > > The reviewer suggested two aerial-only baselines: Shi et al. [75] (CVPR'22) and SliceMatch (CVPR'23). We argued that our aerial-only baseline in Table 1 is conceptually similar to a newer paper, Fervers et al. [21] (CVPR'23 -- code not available at the submission deadline), which performs better than [75] and also SliceMatch on the same or very similar datasets. SliceMatch is a concurrent submission by NeurIPS rules, and code was not publicly available at the submission deadline.
> > >
> > > Moreover, our results show that the gap between aerial-only and aerial+ground-level is too large to be bridged with improvements to the former, which is what motivated our approach to begin with.
> > >
> > > We believe that our baselines are quite thorough: we compare against aerial-only approaches (a variant of our algorithm similar in spirit to [75]), ground-level approaches (structured matching against SfM maps), and semantic maps (OrienterNet).
> > >
> > > **In the visualization results of Figure 4, the representation of pillars and trees on the map is very clear, which is crucial for successful localization and image segmentation. I believe this is mainly due to the utilization of aerial images and street views. On the other hand, OrienterNet only uses semantic maps as input, making it almost impossible to learn the positions of pillars and trees in such maps. Therefore, I think the author should pay more attention to comparing methods that employ aerial images and street views as inputs.**
> > >
> > > This is true of the OpenStreetMap semantic maps that OrienterNet was originally trained with, but the semantic rasters we use in our implementation include both pillars and trees. The full list of categories is the same as in our semantic mapping experiments (see the histograms in Fig. 6): road, building, sidewalk, crosswalk, fence, pole, tree, traffic sign, traffic light, and street light.
> > >
> > > Unlike the ground truth used in those experiments (which is scarce), these rasters are noisy, but they are available everywhere. Qualitatively, they look similar to the examples of Fig. 6 in the paper and Figs. 6 and 7 in the appendix. We would show more examples but we cannot append a PDF to OpenReview during the discussion phase. We will clarify this in the paper and add examples.

---

> > > > ### Comment · Reviewer_PKpW · 2023-08-21
> > > >
> > > > 1. Using the ideas from Fervers et al. [21] to generate a multi-modal neural map is different from the localization work done in [21] itself. The excellent performance of localization in [21] does not imply that the multi-modal neural map generated using the same ideas would perform equally well in localization tasks.
> > > >
> > > > 2. The work by Fervers et al. [21] focuses on multi-camera localization, which is different from the author's experiments on single-camera localization. I believe there is not much comparability between these two methods.
> > > >
> > > > 3. I have checked the release dates of the code, and the author is correct that the work on SliceMatch has not been published. However, I noticed that the work by Shi et al. [75] was published on October 26, 2022, and the author did not provide experimental evidence to demonstrate that their method outperforms Ref[75]. For Table 1, both of them are single-camera localization works, as I mentioned before, the work by Shi et al. [75] relies solely on satellite image for localization.

---

> > > > > ### Author Response · Authors · 2023-08-21
> > > > >
> > > > > **Using the ideas from Fervers et al. [21] to generate a multi-modal neural map is different from the localization work done in [21] itself. The excellent performance of localization in [21] does not imply that the multi-modal neural map generated using the same ideas would perform equally well in localization tasks.**
> > > > >
> > > > > We do not understand this statement. We do not use the ideas of [21], or claim that we build on their work, only that it is closer to the "aerial" variant of our model that we use as a baseline in Table 1. It is also concurrent work (CVPR'23).
> > > > >
> > > > > Regarding the localization process, Fervers et al. [21] perform exhaustive evaluation of the pose probability density function, while we sample high-density modes using RANSAC, which is computationally much more efficient. The two approaches are equivalent in terms of accuracy: they find the pose with maximum likelihood.
> > > > >
> > > > > **The work by Fervers et al. [21] focuses on multi-camera localization, which is different from the author's experiments on single-camera localization. I believe there is not much comparability between these two methods.**
> > > > >
> > > > > Fervers et al. [21] estimates a BEV from an arbitrary number of images, including multi-camera or single-image queries. This is similar to our-ground level encoder (please refer to the sequence-to-sequence alignment examples Sec. C of the appendix). This work is thus very much applicable to our experimental setup and similar to our aerial-only baseline.
> > > > >
> > > > > **I noticed that the work by Shi et al. [75] was published on October 26, 2022, and the author did not provide experimental evidence to demonstrate that their method outperforms Ref[75]. For Table 1, both of them are single-camera localization works, as I mentioned before, the work by Shi et al. [75] relies solely on satellite image for localization.**
> > > > >
> > > > > Shi et al. [75] project features extracted from an overhead view into the ground-level view using ground plane homographies and refine an initial pose. This work assumes that the initial orientation is accurate up to 20 degrees of errors, which is very restrictive. Our experimental setup does not assume any orientation prior so this work is not applicable and cannot be fairly compared to ours.
> > > > >
> > > > > Refining an initial pose as done in [75] is much less robust than explicitly matching a BEV. This is demonstrated by the fact that [75] is outperformed by more recent works, both [21] (Table 1, Ford AV) and OrienterNet [64] (Table 3, KITTI). [75] thus does not represent the state of the art anymore.
> > > > >
> > > > > Lastly, as the reviewer notes, [75] relies on satellite imagery only for mapping. We reiterate our claim that no matter how effective methods such as [21, 75] prove themselves to be, they will still fall short of those that use ground-level imagery to build the maps, as we do, due to occlusions.

---

> ### Comment · Area_Chair_SKKu · 2023-08-18
>
> Dear reviewer PKpW,
>
> could you please tell the authors (and us) whether your concerns have been answered?
>
> Best,
> AC

---

### Official Review · Reviewer_dKqM · 2023-07-03

**Soundness:** 3 good
**Presentation:** 4 excellent
**Contribution:** 3 good
**Rating:** 5
**Confidence:** 4

**Summary:**

This paper proposes a self-supervised learning approach to create a neural representation of the 3D environment from aerial-view and ground-view images. It utilizes contrastive learning to align neural maps based on the same pose and distinguish maps with inconsistent poses. Large-scale pretraining enhances the model's capabilities for visual positioning and semantic mapping. Experimental results on two datasets validate the effectiveness of the approach.

**Strengths:**

1. The overall idea of learning a neural scene representation from multimodal images via pose contrast is both elegant and effective. This approach allows for a comprehensive understanding of scenes by leveraging different modalities, resulting in a robust representation.

2. The implementation of the overall framework is detailed and non-trivial. The monocular inference, multiview fusion, multimodal fusion, and differentiable pose estimation are well executed and contribute together to the final performance.

3. The large-scale pretraining enhances the neural map with high-level semantics. This pretrained model can be further fine-tuned for tasks such as visual positioning and semantic mapping. This versatility makes it a strong candidate for serving as the foundation model in areas like robotic localization and mapping.

4. The writing in the paper is clear and easy to follow. Additionally, the visualization provided is informative.


**Weaknesses:**

1. **Pose supervision**: The authors should discuss the influence of pose accuracy on SNAP's performance and address how reliable pose information can be obtained to improve SNAP's learning capabilities. Exploring the impact of pose accuracy on the model's performance would provide valuable insights.


2. **Overall illustration**: It would be beneficial for the authors to include a pseudo-algorithm to better illustrate the notations used and clarify the training and inference procedures of SNAP. This addition would enhance the understanding of the model's implementation.


3. **Training and inference time**: Given the large scale of the training set, it would be helpful for the authors to discuss the training time of SNAP and justify the necessity of such a large-scale dataset (will reducing the training data significantly reduce the performance of SNAP?). Additionally, considering the practicality for mobile robotics, it would be appropriate to provide information on the inference time on a more affordable GPU. This discussion would provide insights into the computational requirements of the model.


4. **Depth estimation**: The authors should clarify how the depth model handles the large distortions present in the input images. Providing details on how these distortions are addressed and mitigated would enhance the understanding of the model's performance.


5. **Ablation studies**: The reviewer suggests conducting ablation studies to investigate the influence of Equation (1) and the max-pooling used for multimodal fusion. Including these studies would demonstrate the effectiveness of different modules and contribute to a deeper understanding of the model's performance.


6. **Details of semantic mapping**: More information is needed regarding the input, scale, and output of the semantic mapping process. Additionally, since the semantic classes appear to be limited, the authors are encouraged to test the model on publicly available datasets such as SemanticKITTI SSCBenchmark to provide a comprehensive evaluation.


7. **Dataset and code availability**: It is essential for the authors to make the datasets used for pretraining and semantic mapping publicly available. Furthermore, open-sourcing the code and data would greatly benefit the field, allowing researchers to replicate and build upon the presented work.


**Minor**
1. The authors are suggested to show easy, medium, and hard examples’ visualizations respectively.

2. In the literature review section (line 196-line 204), it would be beneficial for the authors to include references to works related to semantic occupancy mapping, specifically addressing voxel representations for large-scale scene understanding (not only just object-centric applications). Adding these related works will provide a broader context. Some relevant works to consider include MonoScene [1], VoxFormer [2], TPVFormer [3], and SemanticKITTI [4].

[1] Cao, A.Q. and de Charette, R., 2022. Monoscene: Monocular 3d semantic scene completion. In Proceedings of the IEEE/CVF Conference on Computer Vision and Pattern Recognition (pp. 3991-4001).

[2] Li, Y., Yu, Z., Choy, C., Xiao, C., Alvarez, J.M., Fidler, S., Feng, C. and Anandkumar, A., 2023. Voxformer: Sparse voxel transformer for camera-based 3d semantic scene completion. In Proceedings of the IEEE/CVF Conference on Computer Vision and Pattern Recognition (pp. 9087-9098).

[3] Huang, Y., Zheng, W., Zhang, Y., Zhou, J. and Lu, J., 2023. Tri-perspective view for vision-based 3d semantic occupancy prediction. In Proceedings of the IEEE/CVF Conference on Computer Vision and Pattern Recognition (pp. 9223-9232).

[4] Behley, J., Garbade, M., Milioto, A., Quenzel, J., Behnke, S., Stachniss, C. and Gall, J., 2019. Semantickitti: A dataset for semantic scene understanding of lidar sequences. In Proceedings of the IEEE/CVF international conference on computer vision (pp. 9297-9307).


**Questions:**

Overall, the paper presents a novel and well-executed approach to learning neural scene representations from multimodal images. Its technical implementation, combined with the potential for practical applications, makes it a valuable contribution to the field. However, there are some concerns listed in the weakness section. This reviewer will consider raising the score if the concerns are well addressed.


**Limitations:**

There seems no checklist.

---

> ### Author Rebuttal · Authors · 2023-08-09
>
> **Exploring the impact of pose accuracy [of the training data] on the model's performance would provide valuable insights.**
>
> We did not try polluting the poses while training. While we can try this, it was not possible to regenerate the data and retrain the model within the rebuttal window.
>
> **It would be beneficial to include a pseudo-algorithm to better illustrate the notations and clarify training/inference procedures**
>
> We will include a glossary and pseudo-algorithm in the appendix for the final version of the paper.
>
> **Training and inference time: [...] it would be helpful to discuss the training time and justify the necessity of such a large-scale dataset (will reducing the training data significantly reduce the performance?)**
>
> The supplementary material mentions that the small and large models were trained for 400k and 200k iterations respectively. This corresponds to 3-4 days of training for each. We anneal the learning rate at half of the total iterations but observe that the loss is still decreasing. Training for a longer time would likely improve the results further but with diminishing returns and prohibitive costs (each model is trained with 16 A100 GPUS). This limitation is similar to other foundation models, like large language models.
>
> The diversity of the training data (i.e. covering multiple neighborhoods of multiple cities) is critical to the generalization of SNAP to different locations. Scaling an earlier version of the model showed that adding more cities (up to 11, as eventually used) directly improves the performance on a held-out test set. We are not able to reproduce such experiments with the final model within the rebuttal period, but will try to add this study to the final version.
>
> **Considering the practicality for mobile robotics, it would be appropriate to provide information on the inference time on a more affordable GPU.**
>
> Currently we only have access to GPUs mounted on remote cluster machines (e.g. A100/P100/V100). We will try to run this on embedded devices and answer this question in the final version, but it is not feasible within the rebuttal window. We will also release the code.
>
> **Depth estimation: The authors should clarify how the depth model handles the large distortions present in the input images.**
>
> The projection model accounts for radial distortion and rolling shutter. Note that the rolling shutter prevents us from undistorting the images, so we feed them as-is to the ResNet backbone. The model learns to handle such distortions given the large amount of data.
>
> **[Ablation] The reviewer suggests conducting ablation studies to investigate the influence of Eq. (1) and the max-pooling used for multimodal fusion.**
>
> **On Eq. 1:** Table 1 in the supplementary material compares SNAP to different variants (of Eq. 1) that either remove the monocular prior (uniform instead of weighted average and variance) or replace it with a more expensive ray conditioning [74]. It is unclear what other aspects of Eq. 1 should be studied: any clarification is very much welcome.
>
> **On max-pooling:** The _max_ operator has a number of theoretical benefits over _average_ or _attention_ pooling:
> - It is transitive (when fusing 3 maps recursively, one can perform pairwise fusion in any order with no impact on the final result)
> - It is robust to noise, as it can ignore inputs with low information (e.g. because of occlusion).
>
> While we validated this experimentally with an earlier version of the model (and thus chose the max), we are not able to reproduce such results with the final model within the rebuttal period, but will try to add this ablation to the final version.
>
> **Details of semantic mapping: More information is needed regarding the input, scale, and output of the semantic mapping process. Additionally [...] the authors are encouraged to test the model on publicly available datasets**
>
> We have 3k non-overlapping, labeled scenes, and use 2k for training and the remaining 1k for testing. The scenes come from 84 different cities across the world, and each of them is represented in the training and testing splits. We will add this information to the revised supplementary material.
>
> As stated in the global response, we will open-source and release the code. While we plan to train and evaluate our model on open benchmarks, we can not do so within the scope of the rebuttal.
>
> **Dataset and code availability: It is essential for the authors to make the datasets used for pretraining and semantic mapping publicly available. Furthermore, open-sourcing the code and data would greatly benefit the field**
>
> As stated in the global response, we will open-source the code. We cannot make the data available because we do not own it. Please note that a large-scale dataset of ground-level images must comply with regulations such as GDPR, including takedown requests, which require non-trivial infrastructure/maintenance and are impossible to enforce after the data has been released. Existing projects that distribute datasets crawled from StreetView do so without permission from the data owners and ignoring these regulations.
>
> **The authors are suggested to show easy, medium, and hard examples’ visualizations respectively.**
>
> We will label each qualitative example accordingly.
>
> **In the literature review section, it would be beneficial for the authors to include references to works related to semantic occupancy mapping, specifically addressing voxel representations for large-scale scene understanding (not only just object-centric applications). Adding these related works will provide a broader context. Some relevant works to consider include MonoScene [1], VoxFormer [2], TPVFormer [3], and SemanticKITTI [4].**
>
> We thank the reviewer for this suggestion. We agree that these works are relevant and will add them to the related work section.
>
> **Limitations: There seems no checklist.**
>
> We note some limitations in L280-282.

---

> > ### Comment · Reviewer_dKqM · 2023-08-22
> >
> > Thank the authors for the response. I will keep my original rating.

---

### Official Review · Reviewer_qSVT · 2023-07-07

**Soundness:** 4 excellent
**Presentation:** 4 excellent
**Contribution:** 4 excellent
**Rating:** 6
**Confidence:** 4

**Summary:**

This paper presents a method for self-supervised learning of BEV features, by using a contrastive loss that links features computed from perspective-view images to features computed from aerial images, where BEV correspondences are given by poses. Interesting method details include depth-weighted splatting in the style of Lift-Splat-Shoot, local decoding/fusion with an MLP, and hard negative sampling via RANSAC. The method provides good localization especially in "hard" cases where the camera moves a lot, and the features are useful pre-training for downstream semantic segmentation tasks.


**Strengths:**

This paper is very well written. The problem and the method are very clearly articulated, the figures and sectioning and style are all helpful to getting the key ideas across efficiently.

Self-supervised feature learning from the combination of car-mounted and aerial data is a new and exciting area, and the approach here is well motivated. The method makes sense.

The approach achieves fairly good results, and outperforms SfM methods in some cases.


Using featuremetric similarity to guide the sampling is a very nice detail.


**Weaknesses:**

The pre-training experiment would be more convincing if it were benchmarked agaginst some existing approach. I am particularly interested to see if the pre-trained model outperforms some SOTA on some BEV segmentation task (e.g., on nuScenes). The danger is something like: as you add more supervised data, the impact of the pre-training goes to zero.

The interpolation operator used in eq 4 is confusing for me. Can you write it or explain it in another way? (One additional question here, beyond the main confusion of that equation, is: are you really doing a plain sum here? Typically a mean makes more sense here, so that the scale of the term does not depend on a hyperparameter like resolution.)

The overall map seems like it may be huge. I need more implementation details on this aspect to really understand what is going on, especially with respect to generating pairs (and non-pairs) for the contrastive loss.


**Questions:**

"Overhead images are ortho-rectified, such that all world points along the z axis project onto the same pixel coordinate."

This sounds odd to me, so I want to double check. I am imagining the aerial imagery is captured from very high up, and the images cover a wide FOV, and only a few pixels will be well-aligned with the z axis (defined as the gravity direction). Fixing this seems like it should require depth information (e.g., especially for tall buildings).


"we define K horizontal planes at heights {zk}, which are uniformly distributed around the camera heights and points of interest"

What exactly are the "points of interest"?


It's interesting that the architecture does not have any BEV CNN involved. To my understanding, it is often this BEV computation which resolves much of the 3D structure of the scene, since depth information is so difficult to estimate from RGB alone (without supervision). Is it really just an MLP applied on each cell independently? The citation here (SimpleRecon) seems not as simple as what's presented here -- it uses a cost volume, various geometric "metadata", and another 2D CNN. Can you comment a bit more here, to help me get a sense of what's going on in the architecture?


"Our approach tightly combines 3D geometry and representation learning instead of relying on expensive Transformer [60] or 3D CNNs [25]."

OK but I think the more reasonable baseline is a 2D CNN applied in BEV (e.g., [26]). This is lightweight also.


"We build mapping segments only from car sequences"

Why? And why discuss the data collected from pedestrians/backpacks, if it is never used?




**Limitations:**

Looks good

---

> ### Author Rebuttal · Authors · 2023-08-09
>
> **I am particularly interested to see if the pre-trained model outperforms some SOTA on some BEV segmentation task (e.g., on nuScenes). The danger is something like: as you add more supervised data, the impact of the pre-training goes to zero.**
>
> Obtaining training labels for 3D semantic tasks is very expensive. Such tasks are also more prone to domain shifts than geometric tasks (like registration) and thus require new labels when deploying models on new environments. Getting enough training data for global generalization (across countries, sensors, times of the day) is unfeasible. Our pre-training leverages camera poses, which are much cheaper to acquire at scale.
>
> The information bottleneck forces SNAP to learn unified representations for objects, like street crossings or lights, that look very different across countries, and would require larger amounts of labels. Showing this on public datasets is difficult since they rarely span more than a few cities (nuScenes has data from Boston and Singapore) and overfit supervised models to local appearance.
>
> **The interpolation operator used in eq 4 is confusing for me.**
>
> We consider a point $ (i,j) $ in the query map, with corresponding query feature $ \bar{\mathbf{M}}\_{ij}^Q\in\mathbb{R}^D $.
>
> We write $ \mathbf{C}\_{ij} $ the $ I\times J $ similarity map between this query feature and each reference feature $ \bar{\mathbf{M}}^R\in\mathbb{R}^{I\times J\times D} $.
>
> This point has a 2D coordinate $ \mathbf{G}\_{ij}^Q $ in the coordinate system of the query grid.
>
> A relative pose candidate $ {}\_R\mathbf{T}\_Q $ transforms this point into the reference coordinate system as $ \mathbf{p} = {}\_R\mathbf{T}\_Q\cdot\mathbf{G}\_{ij}^Q $.
>
> To obtain the similarity of this point given this pose, we bi-linearly interpolate the similarity map at $ \mathbf{p} $: $ \mathbf{C}\_{ij}\left[\mathbf{p}\right] $.
>
> (Line breaks for readability.) We apologize for the confusion and will improve this explanation in the revised paper.
>
> **are you really doing a plain sum here? Typically a mean makes more sense here**
>
> We indeed normalize by the number of query points $I\cdot J$, which is missing in Eq 4. We will fix it in the final version.
>
> **The overall map seems like it may be huge.**
>
> Our neural tiles are 64x16m with 20cm resolution, and 128 dimensions. Our tiles are smaller than SuperPoint (L266): ~1.6 MB vs ~5.3 MB. As stated in the supplementary material (L112), we train with a batch size of 32 over 16 GPUs (=two examples per GPU). We found that reducing the grid to increase the batch size hurt performance, as large areas are necessary to find meaningful hard negatives for the contrastive loss (see appendix, Sec. A).
>
> We will release our code, for reference.
>
> **[On orthorectification of aerial images:] I am imagining the aerial imagery is captured from very high up, and the images cover a wide FOV, and only a few pixels will be well-aligned with the z axis (defined as the gravity direction). Fixing this seems like it should require depth information**
>
> Aerial images are captured by planes and are ortho-rectified using a coarse 3D model derived from the same imagery. We use this data because it is available and aligned to StreetView images (we use it with special permission from the data owners). There is no conceptual difference between ortho-rectified aerial images and satellite imagery (resolution is typically a bit lower for the latter).
>
> **"we define K horizontal planes at heights ${z\_k}$, uniformly distributed around the camera heights and points of interest". What exactly are the "points of interest"?**
>
> We meant uniformly within a range "of interest". We use 32 height planes distributed every 20cm within 12m (L230), starting 4m below the camera. We will clarify this.
>
> **It's interesting that the architecture does not have any BEV CNN involved. [...] it is often this BEV computation which resolves much of the 3D structure of the scene [...] Is it really just an MLP applied on each cell independently? [SimpleRecon] seems not as simple [as this]**
>
> An MLP is indeed applied to each 3D point individually (Eq 2). Its output $\mathbf{X}\_k$ is pooled vertically to obtain cell feature $\mathbf{M}\_{ij}$. We found that the supervision signal and amount of data are sufficient to learn very good monocular depth (appendix, Fig. 3). This, combined with multi-view fusion, makes it unnecessary to apply a top-down BEV CNN or an elaborate vertical pooling mechanism.
>
> Using a simple MLP has multiple benefits: 1) we can infer a map feature for any 2D point and cells, so they don't need to be distributed on a regular grid; 2) neural maps are fully equivariant to any translation/rotation of the coordinate system of the camera poses; 3) avoids boundary artifacts introduced by padding in a BEV CNN.
>
> This idea is similar to SimpleRecon, which replaces an expensive 3D CNN with a simpler per-cell MLP, which can be very effective if fed the right information (here multi-view features and monocular scores). We do agree that the comparison is limited.
>
> **"Our approach tightly combines 3D geometry and representation learning instead of relying on expensive Transformer or 3D CNNs."
> I think the more reasonable baseline is a 2D CNN applied in BEV. This is lightweight also.**
>
> As mentioned above, a BEV CNN easily introduces boundary artifacts due to padding and breaks the rotation equivariance of the mapping process. This CNN would increase the complexity of the model and it is unclear what problem it would solve. Our experiments show that the MLP-only design already performs well.
>
> **Why discuss the data collected from pedestrians/backpacks if it is never used?**
>
> We use both (L215-219): we build neural maps from car images, but use car and backpack images for queries. This mirrors the typical use-case where one would map from car-mounted rigs (StreetView cars) and query with single images (e.g. pedestrians using cell phones). Fig. 2 in the appendix breaks the results down.

---

> > ### Comment · Reviewer_qSVT · 2023-08-12
> > **OK**
> >
> > **Obtaining training labels for 3D semantic tasks is very expensive. [...] Getting enough training data for global generalization (across countries, sensors, times of the day) is unfeasible**
> >
> > This doesn't make much sense to me. NuScenes is already labelled. There's also KITTI and Waymo. Running the experiment only costs GPU time. Since as you point out, NuScenes is not very large compared to StreetView, and especially considering the compute capacity ("each model is trained with 16 A100 GPUs"), it seems like the test might only take a couple hours.
> >
> > **Aerial images are captured by planes and are ortho-rectified using a coarse 3D model derived from the same imagery. [...] There is no conceptual difference between ortho-rectified aerial images and satellite imagery**
> >
> > This still sounds odd. My specific complaint was the claim that *"all world points along the z axis project onto the same pixel coordinate"*. Does the 3D model include buildings, street signs, and trees? Figure 4 suggests not: we can see street poles at an oblique angle, instead of top-down like we should expect if the gravity axis is aligned with the camera axis.
> >
> > **a BEV CNN easily introduces boundary artifacts due to padding and breaks the rotation equivariance of the mapping process. This CNN would increase the complexity of the model and it is unclear what problem it would solve. Our experiments show that the MLP-only design already performs well.**
> >
> > This seems missing the point. It was claimed that the approach is an alternative to "expensive Transformer or 3D CNNs", but those architectures are not the right comparison. The current popular approach is BEV CNNs, which are much cheaper and work well. It may in fact be necessary to quantify this comparison -- is the per-cell MLP really cheaper than a 2D BEV CNN? In any case, the 2D issues pointed out here seem inaccurate or trivial to resolve. Problems with boundary artifacts can be resolved with minor architecture changes (e.g., alternate padding schemes), and anyway I don't think this is a common issue for SOTA; rotation equivariance can be approximated by training at many rotations (as SOTA methods actually do already). Anyway I didn't realize rotation equivariance was a claim of the work -- those words never appear in the paper. Are you planning to add some discussion on this?
> >
> > The rebuttal also missed the very first point of my review: "The pre-training experiment would be more convincing if it were benchmarked against some existing approach."

---

> > > ### Author Response · Authors · 2023-08-16
> > >
> > > **The rebuttal also missed the very first point of my review: "The pre-training experiment would be more convincing if it were benchmarked against some existing approach."**
> > >
> > > We apologize for the confusion, we thought this was covered by the first part of our response. Please see a detailed answer below.
> > >
> > > **This doesn't make much sense to me. [NuScenes/KITTI/Waymo] are already labeled. Running the experiment only costs GPU time. Since as you point out, NuScenes is not very large compared to StreetView [...] it seems like the test might only take a couple hours.**
> > >
> > > We understand that the reviewer would like to know the impact of pre-training when sufficiently large amounts of labeled data are available.
> > >
> > > First, we described the _motivation_ behind our pretraining experiments, which are based on the fact that it's simply _not feasible_ to label data everywhere. We argue that through its bottleneck, our model learns unified representations for "objects" that vary from country to country. A reasonable baseline for this evaluation would be a pre-training method trained only with image poses, but we are not aware of any.
> > >
> > > Second, we tried to answer the reviewer's question by arguing that the results of the experiment they suggest would not be representative, because datasets like NuScenes, KITTI, and Waymo cover only a small geographic area (1 or 2 cities at most), with identical camera setups and models. When evaluating the performance on a given dataset, pre-training with data from the same dataset of course works better than pre-training with data from diverse sources, even though the resulting model will likely generalize poorly to different datasets. We thus expect our pre-training on StreetView (with various cities, viewpoints, camera models) to not perform better than dataset-specific pre-training, but to generalize better overall. As far as we are aware, past works do not study the generalization of BEV semantic prediction across different datasets. Finally, there is a large domain gap between datasets (e.g. KITTI has only front-facing cameras).
> > >
> > > **Does the 3D model include buildings, street signs, and trees? Fig. 4 suggests not: we can see street poles at an oblique angle, instead of top-down like we should expect if the gravity axis is aligned with the camera axis**
> > >
> > > Please note that ortho-rectification of aerial and satellite imagery is a well-known, standard process in GIS applications. Overlapping views (from one or more cameras) are used to correct the distortions inherent to perspective geometry by building an elevation model using stereo (or a known elevation model). Given the small baseline distance and image resolution, fine-grained structures like poles are not resolved in the coarse 3D models. Given that most images are fronto-parallel, this yields only minor artifacts with shifts of only a few pixels. Unfortunately we are prevented from providing external links in the comments, but we will refer the reader to further resources in the final version.
> > >
> > > The poles visible in Fig. 4 are mostly shadows cast by the sun. Note that the shadows are not visible in the map/query StreetView images because they were taken at different times. Using the original aerial data would be much more complicated with little benefit, since the learning model (overhead CNN) can account for small inaccuracies of the rectification by detecting ground features (e.g. base of the poles).
> > >
> > > While we will certainly discuss this in more detail in the final paper, we would like to stress that we use this data simply because we have access to it. The resulting images are very similar to satellite imagery -- which is also often orthorectified.
> > >
> > > **The current popular approach is BEV CNNs, which are much cheaper and work well.**
> > >
> > > A 2D CNN in BEV space can improve an existing BEV feature map, but does not itself perform the image-to-BEV estimation. Existing works rely on attention along polar rays [60], attention between image and BEV tokens [40], lifting and splatting [55], or 3D-to-image projection [26]. Some of them do include BEV CNNs to improve the BEV features but rely on different mechanisms to create an initial BEV map in the first place. These approaches can fuse multi-view information only in BEV space, which requires strong monocular cues and is thus much less accurate than 3D fusion as done in the field of MVS. MVS approaches [97] have often relied on an expensive 3D CNN, mostly because they lacked monocular priors, while our model is sufficiently expressive without it.
> > >
> > > Adding the BEV CNN on top of our model would be more expensive, with likely little benefits. We emphasize in the paper that large batch and map sizes are critical to the performance of the model, and adding a BEV CNN would reduce both.
> > >
> > > **I didn't realize rotation equivariance was a claim of the work**
> > >
> > > It is not a central claim to our work but a side-benefit that is not critical to our main contributions. We will briefly discuss it in the final version.

---

> > > > ### Comment · Reviewer_qSVT · 2023-08-17
> > > >
> > > > > Please note that ortho-rectification of aerial and satellite imagery is a well-known, standard process in GIS applications.
> > > >
> > > > Thank you for the note. My specific complaint the claim that "all world points along the z axis project onto the same pixel coordinate". As you mention, "fine-grained structures like poles are not resolved in the coarse 3D models", so it seems clear that the claim is not true. Am I still missing something?
> > > >
> > > > > The poles visible in Fig. 4 are mostly shadows cast by the sun.
> > > >
> > > > The big white sign seen from an angle in Figure 4c is definitely not a shadow. The shadow is next to it, on the left.
> > > >
> > > > > A 2D CNN in BEV space can improve an existing BEV feature map, but does not itself perform the image-to-BEV estimation.
> > > >
> > > > To me this again seems to be missing the point, but I may have misunderstood something along the way. To my understanding, the MLP is fusing multiview information, just like a 2D BEV CNN would. In both setups, there is typically a separate geometry-based module which corresponds BEV cells to camera pixels. My new impression is that the authors have a lot of knowledge on related work in MVS (where methods keep around the height dimension), and therefore 3D CNNs and Transformers are top of mind, and maybe this is why the claim was made. In BEV detection/segmentation and even in detection from LiDAR, squashing the vertical dimension is actually very common.
> > > >
> > > > > Adding the BEV CNN on top of our model would be more expensive, with likely little benefits.
> > > >
> > > > I agree, but anyway I never suggested this.
> > > >
> > > > > We emphasize in the paper that large batch and map sizes are critical to the performance of the model, and adding a BEV CNN would reduce both.
> > > >
> > > > I doubt the BEV CNN would necessarily reduce batch size. How big is the MLP? A per-cell MLP is like a 2D CNN with all 1x1 kernels. With bigger kernels and a smaller channel dimensions, maybe you could match the compute, or even arrive at something cheaper. (I noticed that the resnet backbone is quite large -- 353M parameters -- but I could not find a parameter count for the MLP.)
> > > >
> > > > > A reasonable baseline for this evaluation would be a pre-training method trained only with image poses, but we are not aware of any.
> > > >
> > > > It doesn't need to be this setup exactly, just anything operating on these inputs or fewer. For example, these days MAE is a go-to self-supervised baseline in all kinds of domains.
> > > >
> > > > Overall my main complaint is still that the pre-training experiment would be more convincing if it were benchmarked against some existing approach. This seems in alignment with PKpW's complaint that the only non-SFM baseline is OrienterNet.

---

> > > > > ### Author Response · Authors · 2023-08-18
> > > > >
> > > > > **My specific complaint the claim that "all world points along the z axis project onto the same pixel coordinate". As you mention, "fine-grained structures like poles are not resolved in the coarse 3D models", so it seems clear that the claim is not true.**
> > > > >
> > > > > Fair enough: that is the goal of the ortho-rectification process, but we concede that it does not always hold. We will amend this statement in the final version, and add more details about the ortho-rectification process to the appendix.
> > > > >
> > > > > **To me this again seems to be missing the point, but I may have misunderstood something along the way. To my understanding, the MLP is fusing multiview information, just like a 2D BEV CNN would. In both setups, there is typically a separate geometry-based module which corresponds BEV cells to camera pixels. My new impression is that the authors have a lot of knowledge on related work in MVS (where methods keep around the height dimension), and therefore 3D CNNs and Transformers are top of mind, and maybe this is why the claim was made. In BEV detection/segmentation and even in detection from LiDAR, squashing the vertical dimension is actually very common.**
> > > > >
> > > > > The point originally raised by the reviewer was that we position our method against "expensive Transformer [60] or 3D CNNs [25]" (L124), and argued that 2D BEV methods are a better comparison. We discuss BEV representations in L198+, and [60] is a BEV method. We concede that this statement can be misleading and will amend L124 to note that the current paradigm has shifted towards squashing the vertical dimension into a BEV.
> > > > >
> > > > > The reviewer also noted in the review that _"a more reasonable baseline is a 2D CNN applied in BEV (e.g., [26]). This is lightweight also."_  But SimpleBEV [26] combines multi-view information by averaging the 3D volumes, and only afterwards squashes the vertical direction and applies the BEV CNN. This is weaker than fusing in 3D with an MLP and the feature variance, as demonstrated by MVSNet [11], which ablates using mean vs variance (Fig 6b, albeit using the loss rather than an evaluation metric). We can add this baseline to the ablation study in the final paper.
> > > > >
> > > > > **I doubt the BEV CNN would necessarily reduce batch size. How big is the MLP? A per-cell MLP is like a 2D CNN with all 1x1 kernels. With bigger kernels and a smaller channel dimensions, maybe you could match the compute, or even arrive at something cheaper. (I noticed that the resnet backbone is quite large -- 353M parameters -- but I could not find a parameter count for the MLP.)**
> > > > >
> > > > > We agree that it is possible that running the MLP on each 3D voxel is as expensive as a BEV CNN with multiple residual layers.
> > > > >
> > > > > **It doesn't need to be this setup exactly, just anything operating on these inputs or fewer. For example, these days MAE is a go-to self-supervised baseline in all kinds of domains.
> > > > > Overall my main complaint is still that the pre-training experiment would be more convincing if it were benchmarked against some existing approach.**
> > > > >
> > > > > The goal of this experiment was to show that training for localization with pose labels (which are cheap) provides an effective pretraining for semantic tasks (much more expensive). Our results show that fine-tuning our pretrained model with a small dataset works much better than training it from scratch. The reviewer asked if pretraining is actually necessary when large amounts of data are available. While that would be an interesting baseline, we do not have truly large-scale semantic annotations, precisely because they are very expensive.
> > > > >
> > > > > Popular self-supervised approaches like MAE and DINO work on single images and require a Transformer to work well. They supervise only for image-level features but not for depth estimation or multi-view fusion, which would need to be learned during the fine-tuning. We therefore think that MAE is not a relevant baseline. We are happy to evaluate, for the final version, a stronger approach if the reviewer can suggest one, but we are not aware of any (using poses or not).

---

### Official Review · Reviewer_Lqgd · 2023-07-07

**Soundness:** 3 good
**Presentation:** 3 good
**Contribution:** 3 good
**Rating:** 6
**Confidence:** 4

**Summary:**

This work develops neural 2D maps in an self-supervised manner by matching multi-view ground level imagery with aerial images.
To construct a neural map (2D map of features), they encode an image into a feature frustum (similar to lift-splat-shoot), then pool vertically - for multi-view images, they perform a similar fusion method and interpolate between image features at multiple defined heights to construct the map.
For supervision, they use known 3-dof poses to align two neural maps. They use a contrastive loss and mine for negative samples using RANSAC.
They train on a large scale streetview dataset with aerial imagery and evaluate performance on a visual positioning task - given 36 views, predict the 3-dof pose of where a query image was taken. Results show improvements over traditional SFM and more recent learning based methods.

**Strengths:**

* Innovative and novel idea with real world applications.
* Technical section fairly clear to follow.
* Overall the work is well written and descriptive.
* The multi-view fusion could be applied to other fields (3D object detection)

**Weaknesses:**

* There are many simplifying assumptions (which the authors have provided discussion for) - relative poses which they "claim can easily be obtained", gravity direction, pose in only 3-Dof.
* Quantitative evaluation is not as fair as I would like. Especially when comparing streetview inputs only, the performance improvement is not particularly strong.
* All ablations are in the supplementary. The system is consists of many design choices and it would be useful to present which choices were critical and which did not matter much.
* All baselines are fairly strong in the "Easy" group. I imagine this is actually the most important category for real world applications since the pose estimation would be applied in the loop.

**Questions:**

* Are the neural maps biased towards features that are greater in height? During training with multi-modal inputs, the aerial tiles are biased towards features that are maximum in height (all other stuff is occluded) and so in theory there is no need for the feature encoder to learn anything below the highest cell.
* Line 105: I didn't understand this. I thought lift-splat-shoot normalized the distribution.
* Is the data used public?

**Limitations:**

Limitations are provided (accuracy + method assumptions).
CO2 usage is specified in supplementary.

---

> ### Author Rebuttal · Authors · 2023-08-09
>
> **Quantitative evaluation is not as fair as I would like. Especially when comparing streetview inputs only, the performance improvement is not particularly strong.**
>
> We note that with StreetView inputs only, our model still outperforms SfM+SuperGlue (a strong and popular baseline) on the medium (43.9 / 56.0 vs 38.0 / 44.4) and hard (29.5 / 41.7 vs 13.1 / 16.1) splits, and also on average (41.0 / 53.2 vs 39.2 / 45.2): see Table 1.
>
> **All ablations are in the supplementary. The system consists of many design choices and it would be useful to present which choices were critical and which did not matter much.**
>
> We believe that the ablation study already substantiates our architecture choices: all the choices evaluated in Table 1 of the supplementary material have a large impact on the localization accuracy. Choices which we observed had very little effect were not considered here.
>
> We put the ablation studies section in the supplementary material due to space constraints, but we plan to move it to the main paper in the camera-ready version if the paper is accepted, as NeurIPS grants an extra page then.
>
> **All baselines are fairly strong in the "Easy" group. I imagine this is actually the most important category for real world applications since the pose estimation would be applied in the loop.**
>
> Baselines based on SfM work well in the presence of easily discriminated structure, such as building facades, but often fail if they are occluded or if the patterns are ambiguous or repeated. Our approach can actually _leverage_ structure such as trees or road markings as semantic cues and localize queries beyond the reach of traditional approaches. Figures 9-14 in the supplementary material show examples that the baselines would fail to resolve: these are all from real settings in some of the most populated cities in the world. We thus think that solving examples in the "medium" and "hard" categories is actually _more_ important for many applications in urban environments.
>
> **Are the neural maps biased towards features that are greater in height? During training with multi-modal inputs, the aerial tiles are biased towards features that are maximum in height (all other stuff is occluded) and so in theory there is no need for the feature encoder to learn anything below the highest cell.**
>
> Looking at the RGB visualizations of the neural maps, tree trunks are clearly distinguishable (in pink), but they are usually not visible in the aerial images, because they are occluded by foliage. This shows that neural maps can encode distinctive features at any height. Also, quantitatively, we observe that using only ground-level imagery performs ~46% better than using only aerial imagery due to the loss of information. Combining both input modalities improves the positioning accuracy over ground-level only by ~8%.
>
> **Line 105: I didn't understand this. I thought lift-splat-shoot normalized the distribution.**
>
> Lift-splat-shoot indeed infers a normalized distribution over depths along each ray. Instead, SNAP infers an absolute score $S$ that is normalized across multiple observations of the same point ($w\_k^n$ in Eq. 1). This makes it possible 1) to compare monocular priors between multiple views and 2) to ignore some features, e.g. those that belong to dynamic objects.
>
> **Is the data used public?**
>
> We cannot release the training data, as we do not own it: please see our comment at the end of the global response. We will open-source and release the code upon acceptance.

---

### Official Review · Reviewer_sYmT · 2023-07-07

**Soundness:** 3 good
**Presentation:** 3 good
**Contribution:** 3 good
**Rating:** 7
**Confidence:** 3

**Summary:**

This paper proposes SNAP, a novel architecture to obtain 2D neural maps from street-view and top-down images. The model works by encoding street-view and top-down images separately using corresponding encoders, projecting features to the top-down 2D space, and fusing them via max-pooling to obtain the final features. The paper also proposes to train the model using a contrastive map alignment objective, where a query map is aligned with a reference map at the ground-truth pose, and differentiated from negative poses. The learned representations are evaluated on single-image positioning and semantic mapping applications. The results show strong improvement over prior approaches.

**Strengths:**

* The problem of learning semantic map representations in a self-supervised way from raw images and camera poses is interesting and well-motivated. While the approach shares some technical similarities with prior work in terms of the mapping functions, the large-scale nature of training, self-supervised learning, and use of top-down + street-view images is exciting.
* The proposed approach is sensible, the training methodology is simple yet effective, and is clearly described in the paper.
* The experiments are well designed and show strong improvements over baselines, and demonstrate the semantic quality of the learned representations.



**Weaknesses:**

* L41 - 42 - Will the "50M street view images" dataset be shared publicly for reproducibility?
* L73 - 77 - How reasonable are the assumptions made? E.g., Is the projection function accurate and noise-free in practice? How easy is it to get ortho-rectified overhead images? How are they obtained?
* Clarifications needed
    * L92 - "picks the best estimate ... handles partial observability" -- how exactly is this achieved?
    * L163 - 165 - How is inference performed to generate pose hypotheses? How scalable is this as map sizes grow larger?
    * Figure 6 - The classification setup for the center plot is not clearly described. Is there quantitative evaluation for the segmentation shown in the right?
* Similarity to prior work not discussed. Please compare and contrast.
    * The mapping approach described in  L99 - 101 is very similar to [R1, R2].
    * The objective of learning features by localization is demonstrated in R1, albeit in a classification approach instead of the proposed contrastive learning.


[R1] Henriques, Joao F., and Andrea Vedaldi. "Mapnet: An allocentric spatial memory for mapping environments." proceedings of the IEEE Conference on Computer Vision and Pattern Recognition. 2018.
[R2] Cheng, Ricson, Ziyan Wang, and Katerina Fragkiadaki. "Geometry-aware recurrent neural networks for active visual recognition." Advances in Neural Information Processing Systems 31 (2018).

**Questions:**

Please address the weaknesses listed.

**Limitations:**

Yes, this is discussed well.

---

> ### Author Rebuttal · Authors · 2023-08-09
>
> **Will the "50M street view images" dataset be shared publicly for reproducibility?**
>
> As stated in the global response, we would really like to, but we cannot, because we do not own the data. Additionally, please note that a large-scale dataset of ground-level images must comply with regulations such as GDPR, including takedown requests, which require non-trivial infrastructure and maintenance and are impossible to enforce after the data has been released. Existing projects that distribute datasets crawled from StreetView do so illegally, without permission from the data owners, and do not comply with such regulations.
>
> **Is the projection function accurate and noise-free in practice?**
>
> Yes, and estimating it for different devices/cameras is easy in practice thanks to Structure-from-Motion.
>
> **How easy is it to get ortho-rectified overhead images? How are they obtained?**
>
> Aerial images are captured by planes and are ortho-rectified using a coarse 3D model derived from the same imagery. We use this data because it is readily available and aligned to StreetView images (note that similarly to StreetView images, we use this data with special permission from the owners). There is no conceptual difference between ortho-rectified aerial images and satellite imagery, other than image resolution, which is typically a bit lower for the latter.
>
> **L92 - "picks the best estimate ... handles partial observability" -- how exactly is this achieved?**
>
> The max operator picks the most distinctive features among all observations. If a view doesn't have good information about a point (e.g. because of occlusion), it can be given a low value that can be ignored by the max.
>
> **L163 - 165 - How is inference performed to generate pose hypotheses?**
>
> Like at training time, pose candidates are computed and scored from high-similarity correspondences obtained by exhaustive matching or even (approximate) k nearest-neighbor search. At inference time we additionally refine the pose via exhaustive matching in a grid centered at the highest-score pose: please refer to appendix E for details.
>
> **How scalable is this as map sizes grow larger?**
>
> This would not scale very well if maps became orders of magnitude larger, such as for instance if GPS priors of any quality are not available. This is also the case for any other approach to fine-grained visual positioning, including structure-based relocalization with maps from SfM. The standard in academia and industry is to use a retrieval index to find coarsely-localized candidates (see e.g. [A]) and refine them. This can also be applied to our approach.
>
> [A] "From Coarse to Fine: Robust Hierarchical Localization at Large Scale", Sarlin et al, CVPR 2019.
>
> **Figure 6 - The classification setup for the center plot is not clearly described. Is there quantitative evaluation for the segmentation shown in the right?**
>
> We evaluate the segmentation with classification metrics. We have 3k non-overlapping, labeled scenes, and use 2k for training and the remaining 1k for testing. The scenes come from 84 different cities across the world, and each of them is represented in the training and testing splits. We will add this information to the revised supplementary material.
>
> All results shown in Fig. 6 in the paper are from the test set. In the middle plot, we show the quantitative results aggregated over the test set. The plot shown on the right is a single qualitative example from the test split. Please note that we added a few more examples in Figs. 6 and 7 in the supplementary material.
>
> **The mapping approach described in L99 - 101 is very similar to [MapNet, Geometry-aware recurrent neural networks for active visual recognition].**
>
> Please note that we cite MapNet in the related work section (L186): its localization step relies on exhaustive voting, similarly to [21, 64], which is much more expensive than our inference scheme based on RANSAC (complexity $R N \log N$ vs $N$, for a map with $N$ cells and voting with $R$ rotations). Its mapping step relies on input depth maps, while SNAP learns monocular priors end-to-end. We thank the reviewer for mentioning the other work -- it is indeed relevant and we will cite and discuss it in the final revision of the paper (NeurIPS does not allow paper revisions during the rebuttal phase).

---

> > ### Comment · Reviewer_sYmT · 2023-08-15
> > **Reviewer response to rebuttal**
> >
> > I thank the authors for their response and for addressing the majority of my concerns. I have a couple of comments based on the rebuttal responses. I trust these will be resolved sufficiently, so I will retain my rating at Accept.
> >
> > * My impression post-rebuttal is that the computational cost for inference and its implication for real-world applications is not represented adequately in the main paper (e.g., an A100 GPU for on-device deployment may not be available). This is a crucial component of such a system that needs to work real-time and on-device. Even if it is computed on the cloud, the resource consumption of such a system is critical if it needs to be scaled to billions of users (e.g., a google maps-like application).
> >
> > * For Fig. 6., it is important to add some more details in the main paper. It is okay to allude to the appendix for complete details, but some minimal information + reference to appendix in the main paper would help.

---

> > > ### Author Response · Authors · 2023-08-16
> > >
> > > **My impression post-rebuttal is that the computational cost for inference and its implication for real-world applications is not represented adequately in the main paper (e.g., an A100 GPU for on-device deployment may not be available). This is a crucial component of such a system that needs to work real-time and on-device. Even if it is computed on the cloud, the resource consumption of such a system is critical if it needs to be scaled to billions of users (e.g., a google maps-like application).**
> > >
> > > Please note that our method needs powerful GPUs only for training: as discussed in the paper, it is crucial to build maps over relatively large areas in order to mine difficult negatives for contrastive learning, and this requires memory. Once trained, we could pre-compute the neural maps using other GPUs. At inference time, we would extract features only for the query image: this model is not heavy, and we already experimented with a lighter ResNet-50 encoder (Table 1). The alignment itself is even simpler and requires only interpolation and dot products, but it seems unlikely that the whole localization stack would run on-device, primarily because this would require storing the map locally. The closest comparison is the visual positioning system of [50] (a world-scale, SfM-based localization stack), which uploads query images to the server (Sec. 8.1).
> > >
> > > We agree with the reviewer that there are open questions regarding an eventual deployment, and that building neural maps for the entire world is a significant undertaking, but we do not claim to do this here. It can take months or even years to productize such a system, and we believe it is out of the scope of the paper.
> > >
> > > **For Fig. 6., it is important to add some more details in the main paper. It is okay to allude to the appendix for complete details, but some minimal information + reference to appendix in the main paper would help.**
> > >
> > > We agree, this was an oversight. We will use some of the extra page in the camera-ready version to correct this if the paper is accepted, adding the details discussed in the rebuttal.

---

### Author Rebuttal · Authors · 2023-08-09

We thank the reviewers for their time and well-thought comments. Reviewers noted very clearly that our method is **novel, well-motivated, and exciting**:
* _"the large-scale nature of training, self-supervised learning, and use of top-down + street-view images is exciting"_ -- R.sYmT
* _"the approach here is well motivated. The method makes sense"_ -- R.qSVT
* _"Self-supervised feature learning from the combination of car-mounted and aerial data is a new and exciting area"_ -- R.qSVT.
* _"The overall idea of learning a neural scene representation from multimodal images via pose contrast is both elegant and effective"_ -- R.dKqM
* _"strong candidate for serving as the foundation model in areas like robotic localization and mapping"_ -- R.dKqM
* _"Innovative and novel idea with real world applications."_ -- R.Lqgd

They also praised our **formulation and presentation**:
* _"the training methodology is simple yet effective, and is clearly described in the paper"_ -- R.sYmT
* _"well written and descriptive"_ -- R.Lqgd
* _"very clearly articulated, the figures and sectioning and style are all helpful to getting the key ideas across efficiently"_ -- R.qSVT
* _"The implementation of the overall framework is detailed and non-trivial. The monocular inference, multiview fusion, multimodal fusion, and differentiable pose estimation are well executed and contribute together to the final performance."_ -- R.dKqM

And mostly agree that it **performs strongly,** against both classical and learned approaches:
* _"The results show strong improvement over prior approaches."_ -- R.sYmT
* _"Results show improvements over traditional SFM and more recent learning based methods."_ -- R.Lqgd
* _"The approach achieves fairly good results, and outperforms SfM methods in some cases."_ -- R.qSVT
* _"Experimental results on two datasets validate the effectiveness of the approach."_ -- R.dKqM

We address their questions in separate responses. A common concern was **reproducibility**: we can now commit to releasing the code, as open source (Apache 2.0 license). Note that we cannot release the training data, as we do not own it. Please note that a large-scale dataset of ground-level images must comply with regulations such as GDPR, including takedown requests, which require non-trivial infrastructure and maintenance and are impossible to enforce after the data has been released. Existing projects that distribute datasets crawled from StreetView do so illegally, without permission from the data owners, and do not comply with such regulations.

Another suggestion was evaluation on **open datasets**: we are working on this, along with open-sourcing of the code, but it is a major effort that cannot be completed during the rebuttal period.

---

### Decision · Program_Chairs · 2023-09-21

**Decision:**

Accept (poster)

**Comment:**

This paper describes a method capable of generating 2D maps from two different types of sources, street view images and aerial photographs. This is performed through a contrastive map alignment objective.

The work has received a majority of positive ratings (5,6,7,4,6), but there was also an ongoing discussion with weaknesses raised by reviewer PKpW regarding comparisons with an aerial variant only. This could be addressed by the authors through the rebuttal and discussions. Other issues were simplifying assumptions, limited comparisons,

This paper was flagged for privacy and security concerns, but the authors do address this issue.

The AC judges that the paper provides valuable contributions to the field and recommends acceptance.